# PROMPT SKETCHING FOR LARGE LANGUAGE MODELS

## ABSTRACT

Many recent prompting strategies for large language models (LLMs) query the model multiple times sequentially – first to produce intermediate results and then the final answer. However, using these methods, both decoder and model are unaware of potential follow-up prompts, leading to disconnected and undesirably wordy intermediate responses. In this work, we address this issue by proposing prompt sketching, a new prompting paradigm in which an LLM does not only respond by completing a prompt, but by predicting values for multiple variables in a template. This way, sketching grants users more control over the generation process, e.g., by providing a reasoning framework via intermediate instructions, leading to better overall results. The key idea enabling sketching with existing, autoregressive models is to adapt the decoding procedure to also score follow-up instructions during text generation, thus optimizing overall template likelihood in inference. Our experiments show that in a zero-shot setting, prompt sketching outperforms existing, sequential prompting schemes such as direct asking or chain-of-thought on 7 out of 8 LLM benchmarking tasks, including state tracking, arithmetic reasoning, and general question answering. To facilitate future use, we release a number of generic, yet effective sketches applicable to many tasks, and an open source library called `dclib`, powering our sketch-aware decoders.

## 1 INTRODUCTION

While early prompting strategies for large language models (LLMs) focused on simple trigger phrases to elicit the desired responses (Kojima et al., 2022), more recent work considers conversational (Ouyang et al., 2022), multi-part, and template-guided LLM interactions, where a model is queried several times in a constrained way, based on a template or grammar. This offers control over LLM reasoning by filling in a template of pre-defined steps (Beurer-Kellner et al., 2023; Lundberg and Ribeiro), allows interfacing with automated software systems, and enables syntactically reliable code generation (Poesia et al., 2022).

**Key Challenge: Decoding with (Hard) Structural Constraints** We consider a simple application, illustrated in Figure 1. The goal is to generate a list of items, satisfying two *hard* requirements: (1) the result should be a dashed list of exactly four items and (2) the second item should be *Frisbee*. To guarantee that these requirements are satisfied, prompting and fine-tuning alone are insufficient, as unconstrained LLMs remain inherently stochastic, even with good instructions, demonstrations, or training Arora et al. (2023); Zhao et al. (2021). To address this issue, template-guided inference constructs a template from the (hard) constraints, leaving multiple holes for the LLM to fill during the generation (top right, Figure 1). Unfortunately, the naive strategy of calling an unconstrained model (Figure 1, left) for each placeholder fails frequently, as the model runs on, beyond the template, generating many items per placeholder. A practical alternative is *stop-and-go* inference (middle): By feeding the template incrementally, item-by-item, and enforcing stopping-conditions for each call, we can force the overall output to adhere to the template. While this method is effective for output formatting (Beurer-Kellner et al., 2023; Lundberg and Ribeiro), the model remains unaware of the overall template when decoding each placeholder, leading to sub-optimal reasoning strategies. For instance, in Figure 1, *stop-and-go* generates *Frisbee* as the first item, leading to a repetition of the word, which would otherwise be unlikely under the model's distribution. This example raises three important questions: (1) How does stop-and-go inference compare to unconstrained inference – Beurer-Kellner et al. (2023); Lundberg and Ribeiro do not evaluate this – in terms of overall model performance on reasoning tasks? (2) Can we improve on naive stop-and-go inference by anticipating

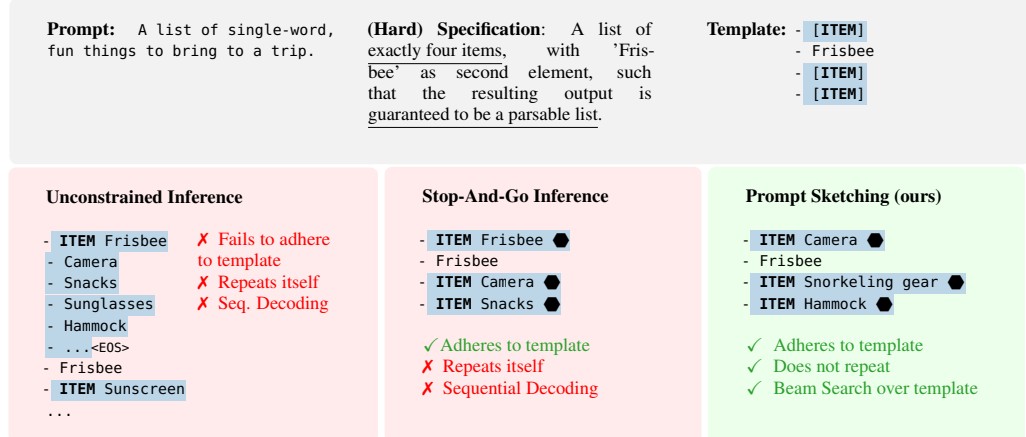

Figure 1: *Prompt Sketching* is a novel inference method for template-guided text generation with LLMs. In comparison to standard inference and sequential stop-and-go inference, prompt sketching optimizes overall template likelihood, prevents repetitions, and adheres to the template structure. Output generated by the model is highlighted, and enforced stopping phrases are indicated as ⬣.

the overall template during generation? And, (3) what are the general effects of these inference methods, i.e., do they impair or improve the model's reasoning capabilities?

**This Work: Prompt Sketching**  To answer these questions, we present *prompt sketching*, a novel framework for template-guided LLM inference. The key technical difference of sketching in contrast to prior techniques is that we phrase the entire template as one segmented sequence decoding problem, rather than multiple isolated model calls. This, (1) theoretically anchors standard stop-and-go inference as a special case and (2) allows us to generalize and implement novel, sketch-aware decoding procedures based on beam search, that optimize templates end-to-end. Figure 1 compares sketch-aware decoding (right) with unconstrained inference (left) and stop-and-go (middle). Sketching allows us to adhere to the provided prompt template, while also optimizing multiple placeholder variables jointly, in this case, avoiding a repetition of *Frisbee*. We carry out an extensive experimental evaluation, showing that sketching outperforms non-templated prompting methods like chain-of-thought on 7/8 LLM reasoning tasks, demonstrating the effectiveness of template-guided inference in general reasoning. In our experiments, sketching allows us to consistently enforce reasoning strategies across all tasks, enabling a more controlled form of LLM programming going beyond simple prompting. For 5/8 tasks, we even observe significant improvements over simple stop-and-go templating, demonstrating that sketch-aware decoding and joint optimization of multiple variables are crucial components of effective template-guided LLM inference.

**Main Contributions**  Our core contributions are:

- A framework of prompt sketching, phrasing multi-step and template-guided LLM inference as a segmented sequence decoding problem.
- Two novel sketch-aware decoding procedures, transferring several insights from constrained sequence decoding to general template-guided inference.
- A collection of ready-to-use, generic prompt sketches that work well with a number of hard LLM reasoning tasks and can be easily adapted.
- The first extensive evaluation of sketching using stop-and-go as well as a several novel (sketch-aware) decoding strategies, along with a comparison to non-templated inference.

Additionally, we publish an open source library `dclib`, included in the supplementary material (see App. E), which allows the implementation of sketch-aware decoding procedures on top of OpenAI, `transformers`, and `llama.cpp` models, thus facilitating future research and open source work.

## 2 BACKGROUND

We first provide relevant background on prompting and decoding, before discussing prompt sketching.

**Decoding** Most recent language models operate left-to-right only, i.e., they predict a probability distribution $p(y_t|\boldsymbol{y}_{<t}, \boldsymbol{x})$ over the next token $y_t$ given an input sequence $\boldsymbol{x} = \langle x_1, x_2, ..., x_n \rangle$ and previously predicted tokens $\boldsymbol{y}_{<t} = \langle y_1, y_2, ..., y_t \rangle$. Thus, a core task is to transduce or decode a model output $\boldsymbol{y}^\star$ that maximizes some scoring function:

$$\boldsymbol{y}^\star = \arg\max_{\boldsymbol{y} \in \mathcal{Y}} \text{score}(\boldsymbol{y}, \boldsymbol{x}). \tag{1}$$

A popular choice for this scoring function is the posterior or joint probability assigned to the decoded sequence by the language model. This leads to the so-called maximum a posteriori (MAP) solution:

$$\boldsymbol{y}_{\text{MAP}} := \arg\max_{\boldsymbol{y} \in \mathcal{Y}} p(\boldsymbol{y}|\boldsymbol{x}) = \arg\max_{\boldsymbol{y} \in \mathcal{Y}} \Pi_{t=1}^N p(y_t|\boldsymbol{y}_{<t}, \boldsymbol{x}) = \arg\max_{\boldsymbol{y} \in \mathcal{Y}} \Sigma_{t=1}^N \log p(y_t|\boldsymbol{y}_{<t}, \boldsymbol{x}) \tag{2}$$

However, solving the MAP decoding exactly is generally intractable, as it requires all conditional probabilities $p(y_t|\boldsymbol{y}_{<t}, \boldsymbol{x})$ over an exponentially large search space to be evaluated. To solve this problem, a range of decoding strategies have been introduced, which aim to find approximate solutions. To discuss them, it is helpful to imagine $\mathcal{Y}$ as a tree with the prompt or prefix $\boldsymbol{x}$ at the root and children of a node corresponding to possible continuations, all scored by $\text{score}(\boldsymbol{y}_{<t}, \boldsymbol{x})$.

**ARGMAX Decoding** corresponds to a depth-first search of our decoding tree that terminates once the first solution has been found. Operationally, at every decoding step, we extend our hypothesis $\boldsymbol{y}_{<(t-1)}$ by choosing the next token $y_t$ to maximize $\text{score}(y_t \mid \boldsymbol{y}_{<(t-1)}, \boldsymbol{x})$:

$$\boldsymbol{y}_{\text{ARGMAX}} := \bigoplus_{t=1}^N \arg\max_{y_t \in \mathcal{Y}} p(y_t|\boldsymbol{y}_{<t}, \boldsymbol{x}) \tag{3}$$

where $\oplus$ denotes concatenation. ARGMAX decoding is efficient, but will also disregard many alternative hypotheses due to its greedy nature.

**Beam Search** corresponds to a breadth-first search in the decoding tree where the breadth (at every tree depth) is limited to the beam width $n$. Operationally, we first determine the $n$ best continuations of all of our $n$ hypotheses and then retain the $n$ best ones across all these $n^2$ continuations. This yields high-quality solutions at moderate computational cost, making Beam Search popular across a wide range of tasks. Interestingly, thus obtained solutions often outperform exact decodings (or very large beam widths) in down-stream tasks (Holtzman et al., 2020). Meister et al. (2020) suggest that this is due to beam search inducing a regularization towards uniform information density, which is preferred in human speech.

**Grid Beam Search** (Hokamp and Liu, 2017) extends beam search to facilitate constraint decoding, i.e., transducing a response such that it contains certain strings or satisfies constraints. As sequences complying with such constraints typically achieve a much lower score than natural model predictions, they would never be included using vanilla beam search. Grid beam search solves this problem by introducing separate comparison pools for hypotheses satisfying different numbers of constraints. To avoid a linear increase in beam width and thus computational cost in the number of constraints, Post and Vilar (2018) introduce a dynamic beam allocation scheme that keeps the total beam width constant and assigns slots on this beam depending on the number of satisfied constraints.

**Length Normalization** (Wu et al., 2016) is frequently employed to compare sequences of different lengths, to compensate for the summation of additional negative logprobs. We can simply weight our scoring function with a length normalization term, parametrized by $\beta \in \mathbb{R}^{\geq 0}$ and $\alpha \in [0, 1]$:

$$w = \frac{(\beta + 1)^\alpha}{(\beta + |\boldsymbol{y}|)^\alpha}, \tag{4}$$

where $\beta = 0$ and $\alpha = 1$ recovers the mean and $\alpha = 0$ no normalization.

## 3 PROMPT SKETCHING

The core of prompt sketching is template-guided LLM inference, i.e., alternating model output with template-derived intermediate tokens. This is different from sequential prompting methods

like *chain-of-thought* or *answer-only*, where first, the model consumes an input such as a question or instructions and then generates an answer in an unconstrained way. More formally, we consider a sketch $\mathcal{S}$ to be a template of the form $\mathcal{S}$ := `"<p₁> [v₂] ... <p_{k-2}> [v_{k-1}] <p_k>"` where, $p_i$ are deterministic sequences of tokens, specified by the template, and $v_i$ are variables that are completed by the model. This definition captures existing forms of prompting, where e.g. *answer-only* (AO) can be written as $\mathcal{S}_{AO}$ := `"<Q> A: [ANSWER]"` and *chain-of-thought* (CoT) prompting as $\mathcal{S}_{CoT}$ := `"<Q> A: Let's think step by step. [COT]."`, where `<Q>` corresponds to a question and the variable `COT` contains model reasoning as well as the final answer.

**Single vs. Multi-Variable Sketches**   We consider $\mathcal{S}_{AO}$ and $\mathcal{S}_{CoT}$ as sequential, single-variable sketches, as the variable is placed at the end of the template. The model, therefore, first digests all provided information such as a question and reasoning instructions before generating the answer. In contrast, with more general sketches, values for multiple variables can be generated, and deterministic intermediate instructions can be inserted during the generation process. Existing examples of multi-variable problems include conversational systems like ChatGPT, agentic prompting like ReAct (Yao et al., 2022a), language model programming (Beurer-Kellner et al., 2023), and language model cascades (Dohan et al., 2022).

**Autoregressive Sketch Decoding**   Sketching extends the range of decoding strategies beyond just sequential generation. However, most language models are still simple next-token predictors, i.e., given some prompt $\boldsymbol{x}$, they generate a sequence of tokens $\boldsymbol{y}$ autoregressively, that is, one token at a time, conditioned only on the previously generated tokens:

$$p(\boldsymbol{y}|\boldsymbol{x}) = \prod_{i=1}^{|y|} p(y_i|\boldsymbol{x}, y_{<i}) \tag{5}$$

To align this with sketching, we split the generated sequence $\boldsymbol{y} = \{y_1, \ldots, y_n\}$, including both deterministic and variable portions, into $k$ consecutive chunks $\mathcal{C}_{\boldsymbol{y}} = \{\boldsymbol{c}_1, \ldots, \boldsymbol{c}_k\}$ of lengths $n_1, \ldots, n_k$ respectively, i.e., $\mathcal{C}_{\boldsymbol{y}} = \{\{y_1, \ldots, y_{n_1}\}, \ldots, \{y_{n_{(k-1)}+1}, \ldots, y_{n_k}\}\}$. Each chunk in $\mathcal{C}_{\boldsymbol{y}}$ is then associated either with a deterministic prompt part $p_i$ or a model-predicted variable $v_i$. The overall joint probability of all chunks is then defined as

$$p(\boldsymbol{c}_1, \ldots, \boldsymbol{c}_k) = \prod_{j=1}^{k} \prod_{i=n_{j-1}+1}^{n_j} p(y_i|y_{<i}) \tag{6}$$

Crucially, we derive the values of all chunks from a single sequence of tokens $\boldsymbol{y}$, which can be predicted sequentially using an autoregressive model. A chunk- and variable-partitioned sequence can then be leveraged by decoding algorithms to obtain higher-quality responses or inject deterministic phrases during generation. The main challenge of this approach is the chunking strategy, i.e., a way to split a generated sequence of tokens $\boldsymbol{y}$ into chunks of an a priori unknown length to determine which variables they should be associated with.

**Chunking with Stopping Phrases**   Like in stop-and-go inference, sketching relies on the use of per-variable stopping phrases (SPs). SPs are specified as part of a prompt sketch and terminate the generation of the current chunk $i$ on occurence. This allows us to chunk the output sequence $\boldsymbol{y}$, assigning the resulting subsequences to variables $v_i$, and keep the model from running-on without respecting the sketch template. In case no specified stopping phrase occurs before the model predicts its designated *end-of-sequence* token, we do not terminate the entire generation process, but only the decoding of the current chunk, unless no further chunks remain.

**Deterministic and Constrained Chunks**   To inject deterministic phrases during generation, we force a predetermined sequence $p_i$ to be decoded, while still evaluating its likelihood $p(\boldsymbol{c}_i|\boldsymbol{c}_{<i})$. Further, we consider *constrained variables* as a special case of non-deterministic variables, whose values are predicted by the model, but can only be chosen from a restricted set of sequences (e.g., only numbers, matching a regular expression, etc.). To implement constrained variables, we rely on the LMQL query language for LLMs (Beurer-Kellner et al., 2023). This allows us to mask out all tokens that will not satisfy a given constraint during generation, such that the resulting value of some restricted variable $c_i$ is guaranteed to satisfy the constraint.

**Example** We show two example sketch templates in Figure 2. In the Reasoning Framework example, we guide the model's reasoning process by inserting deterministic phrases such as `"On the one hand"`, `"On the other hand"`, or `"In conclusion"` inbetween generated reasoning steps. In the Interleaved Reasoning example, we feed the model our problem definition, e.g. sentence by sentence as chunks $Q_i$, prompting for intermediate results after each one. Once the full problem description has been fed to the model, we generate the overall conclusion and answer.

### 3.1 SKETCH-AWARE DECODING

Sketching allows us to denote template-guided LLM inference as one long, segmented sequence decoding problem. With greedy ARGMAX decoding and autoregressive model conditioned on previously generated tokens only, this recovers stop-and-go inference. As discussed in Section 1, however, this form of sequential decoding does not account for yet-to-come parts of the template. At the same time, we operate greedily so after a deterministic chunk has been inserted, we cannot retroactively change the previously generated variable values.

**Reasoning Framework**
\<Question\>
Let's weigh our options:
On the one hand `[THOUGHT]`
However, on the other
hand `[THOUGH]`
In conclusion, `[ANSWER]`

**Interleaved Reasoning**
Let's digest this step by step.
\<Q₁\>
So now, `[STATE]`
\<Q₂\>
So now, `[STATE]`
Overall this means `[ANSWER]`

Figure 2: Two examples of simple multivariable sketches.

To address this, we leverage the probabilistic understanding of sketching and propose a novel class of decoding procedures that, in contrast to traditional token-level decoders, operate on the level of the template to guide the decoding process end-to-end. Concretely, we experiment with two novel decoder adaptations, namely: (1) Hierarchical Variable-Level Beam Search (VAR) and (2) Grid-Based Beam Search (BEAMVAR). Next, we discuss the implementation of these methods in more detail.

**VAR: Variable-Level Beam Search** is based on the idea of applying beam search on the level of the decoded placeholder variables. This means that instead of extending each active hypothesis by the $n$ most likely next tokens, we extend it by $n$ sampled values for the currently decoded variable. Starting with an empty sequence of tokens, we decode variable by variable. When at variable $v_i$, we have at most $n$ hypotheses for which the variables $v_{<i}$ have been chosen. For each of them, we then generate $n$ proposals for variable $v_i$, thus giving us $n^2$ hypotheses over the variables $v_{\leq i}$. Among these, we then select the $n$ most likely ones according to the model score and move to the next variable. Deterministic chunks are handled by appending them to the set of active hypotheses all at once. This process is repeated until all variables have been decoded. See App. A, for a pseudo-code implementation of VAR.

**BEAMVAR: Variable-Grid Beam Search** is based on the idea that the number of decoded variables is an important measure of decoding progress and should thus be considered when comparing the scores of different sequences during token-level beam search, to decide which to explore further. This is particularly important in the presence of deterministic chunks, which, by their very nature, typically have lower likelihoods under the model distribution than non-deterministic variables and would thus never be included in a decoded hypothesis. To this end, we adapt the dynamic beam allocation method of Post and Vilar (2018) to the sketching setting and propose Variable-Grid Beam Search (BEAMVAR): We partition our beam width into separate pools depending on the currently decoded variable $v_i$ and only compare scores per pool. To decide how many slots to allocate to each pool and thus variable, we divide the beam width by the number of unique, currently decoded variables and allocate the remainder to the pool with the most decoded variables, reassigning unused slots to pools decoding later variables, to ensure progress at the template-level. A pseudo-code implementation of BEAMVAR can be found in App. A.

## 4 EXPERIMENTAL EVALUATION

We focus our evaluation on the following questions: (1) Is templated-guided inference and sketching effective at improving the performance of LLMs on reasoning tasks? (2) Can sketch-aware decoders outperform existing decoders in and outside of the sketching setting? And (3), what kind of tasks benefit the most from sketching? To answer these questions, we compare model performance with non-templated, sequential inference on a wide range of different reasoning benchmarks for LLMs (Section 4.1) and also experiment with novel applications enabled by prompt sketching (Section 4.2).

Table 1: `text-davinci-003` task accuracy with *Answer-Only*, *Chain-Of-Thought*, and *Prompt Sketching* (ours) using ARGMAX, BEAM, BEAMVAR (ours) and VAR (ours) decoding. Each configuration is evaluated on 100 uniformly sampled instances per task. Best results are bold.

| Task | Sequential Decoding | | | | | Sketch-Aware Decoding (ours) | | | |
| --- | --- | --- | --- | --- | --- | --- | --- | --- | --- |
| | *Answer-Only* | *Chain-Of-Thought Prompting* | | | | *Prompt Sketching (ours)* | | | |
| | ARGMAX | ARGMAX | BEAM | **BEAMVAR** | VAR | ARGMAX | **BEAMVAR** | VAR | BEAM |
| *Logical Reasoning* | | | | | | | | | |
| Date Understanding[+] | 0.48 | 0.70 | **0.75** | **0.75** | 0.69 | 0.72 | 0.73 | 0.66 | - |
| Information Essentiality[**] | - | - | - | - | - | 0.01 | **0.25** | 0.06 | 0.15 |
| *Question Answering* | | | | | | | | | |
| AQuA (Ling et al., 2017) | 0.31 | 0.37 | 0.37 | 0.35 | 0.35 | 0.40 | **0.47** | 0.35 | - |
| StrategyQA (Geva et al., 2021) | 0.68 | 0.71 | 0.72 | 0.67 | 0.67 | 0.69 | **0.77** | 0.66 | - |
| *Arithmetic Reasoning* | | | | | | | | | |
| Multistep Arithmetic[+] | 0.20 | 0.43 | 0.44 | **0.49** | 0.44 | 0.45 | 0.48 | 0.38 | - |
| GSM8K (Cobbe et al., 2021) | 0.08 | 0.56 | 0.58 | **0.64** | 0.57 | 0.57 | 0.53 | 0.59 | - |
| *Interleaved Reasoning* | | | | | | | | | |
| Tracking Shuffled Objects[+] | 0.19 | 0.62 | 0.47 | 0.52 | 0.62 | 0.64 | 0.62 | **0.66** | - |
| Matrix Shapes[+] | 0.61 | 0.77 | 0.77 | 0.71 | 0.76 | 0.81 | 0.79 | **0.85** | - |

[+] Tasks extracted from the BIG benchmark suite (Srivastava et al., 2022).
[**] Specifically adapted for our evaluation of forward referencing prompts.

**Models**   We use OpenAI's `text-davinci-003` InstructGPT model (175B parameters; Ouyang et al. (2022)) and *Llama-2 Chat* (13B parameters; Llama-2 in the following; Touvron et al. (2023)) to evaluate. While `text-davinci-003` clearly is the more capable model, we find that Llama-2 provides an interesting comparison point for the applicability of sketching to smaller, more open models. We further also experimented with OpenAI's smaller `text-curie-001` model (comparative study in App. C.2).

**Baselines**   As a baseline, we compare sketching to non-templated zero-shot formulations of *answer-only* ($AO$) and *chain-of-thought* ($CoT$), using *zero-shot CoT* (Kojima et al., 2022) for the latter. Examples of all used prompts/sketches are given in App. F. During generation, no task demonstrations are provided and the model is prompted with simple instructions only. This highlights a core benefit of sketching: the ability to precisely guide the model during generation without concrete demonstrations. Still, we also include a comparison with few-shot prompting in App. C.1, which is generally orthogonal to sketching.

**Sketched Chain-Of-Thought**

```
"Q: <question>"
"Answer Choices: (A)...(B)..."
"A: Let's think step by step."

for i in range(12):
 "- [THOUGHT]"
 if not THOUGHT.endswith("\\n"):
 "\\n"
 if "answer" in THOUGHT: break

"Overall this means,[CONCLUSION]
 Therefore, among A through E,
 the answer is[answer]"
```

Figure 3: A sketched formulation of *chain-of-though* reasoning.

**Datasets and Sketches**   We evaluate on a total of 8 LLM reasoning tasks. For each task, we apply one of two generic sketch templates: For arithmetic and logical reasoning, date understanding, and general question answering, we rely on a sketched form of *chain-of-thought*, as shown in Figure 3. For state tracking and matrix shape reasoning, we employ an *interleaved reasoning* sketch, as shown in Figure 2, splitting task descriptions into sentences and interleaving them with the model's reasoning steps. For a detailed description of the tasks and sketches, we refer to App. F.

**Compute and Dataset Size**   The total costs of our OpenAI experiments are roughly $4,000 USD in API use. To limit these costs for our OpenAI experiments specifically, we evaluate only 100 uniformly random samples per task-decoder configuration, with confidence bounds reported in App. C.3. For Llama-2, on the other hand, we run all of our experiments on 1000 samples per task (or the full datasets), using a single NVIDIA H100 GPU with 80GB memory.

**Decoder Configurations**   As a baseline for our sketch-aware decoding procedures, we compare with ARGMAX and traditional beam search (BEAM), applied to each sketch variable individually. Based on this, we examine the benefit of sketching with and without our sketch-aware decoders VAR and BEAMVAR. For BEAM, VAR, and BEAMVAR we use a beam width of $n = 2$ and rely on length normalized scoring in line with previous work (Wu et al., 2016), using $\beta = 0$ and $\alpha = 0.7$.

Table 2: Evaluation results for Llama-2 Chat (13 billion parameters) analogous to Table 1. Zero-shot task accuracy with *Answer-Only*, *Chain-Of-Thought*, and *Prompt Sketching* (ours) using ARGMAX, BEAM, BEAMVAR (ours) and VAR (ours) decoding. Best results in bold.

| Task | Sequential Decoding | | | | | Sketch-Aware Decoding (ours) | | | |
|---|---|---|---|---|---|---|---|---|---|
| | *Answer-Only* | *Chain-Of-Thought Prompting* | | | | *Prompt Sketching (ours)* | | | |
| | ARGMAX | ARGMAX | BEAM | BEAMVAR | VAR | ARGMAX | BEAMVAR | VAR | BEAM |
| *Logical Reasoning* | | | | | | | | | |
| Date Understanding[+] | 0.496 | 0.591 | 0.599 | 0.613 | 0.580 | 0.634 | **0.656** | 0.642 | - |
| Information Essentiality[**] | - | - | - | - | - | 0.088 | **0.132** | 0.132 | 0.132 |
| *Question Answering* | | | | | | | | | |
| AQuA (Ling et al., 2017) | 0.231 | 0.291 | **0.311** | 0.275 | 0.271 | 0.239 | 0.255 | 0.243 | - |
| StrategyQA (Geva et al., 2021) | 0.564 | 0.555 | 0.566 | 0.570 | 0.568 | 0.638 | 0.630 | **0.640** | - |
| *Arithmetic Reasoning* | | | | | | | | | |
| Multistep Arithmetic[+] | 0.038 | 0.133 | 0.120 | 0.138 | 0.132 | 0.126 | **0.142** | 0.103 | - |
| GSM8K (Cobbe et al., 2021) | 0.049 | 0.276 | 0.296 | 0.305 | 0.296 | 0.320 | **0.353** | 0.350 | - |
| *Interleaved Reasoning* | | | | | | | | | |
| Tracking Shuffled Objects[+] | 0.197 | 0.196 | 0.204 | 0.198 | 0.188 | 0.227 | 0.210 | **0.234** | - |
| Matrix Shapes[+] | **0.227** | 0.068 | 0.065 | 0.056 | 0.191 | 0.205 | 0.200 | 0.204 | - |

[+] Tasks extracted from the BIG benchmark suite (Srivastava et al., 2022).
[**] Specifically adapted for our evaluation of forward referencing prompts.

## 4.1 TASK ACCURACY

In Tables 1 and 2, we report our main results on task performance with `text-davinci-003` and Llama-2, respectively. Considering only ARGMAX decoding, we consistently observe improved or maintained performance with sketching, as compared to sequential $CoT$ or $AO$ (7 out of 8 improved for `text-davinci-003`, 6 out of 8 with Llama-2). This shows, that simple ARGMAX sketching can already be effective (up to 4% and 8% points improvement for `text-davinci-003` and Llama-2 respectively). Manual inspection reveals that sketching consistently results in clearly structured reasoning, while with $CoT$ the model makes a seemingly random choice about the form of reasoning applied to each sample (simple text, a list of steps, etc.), impairing task accuracy (see App. F for detailed examples).

**Llama-2** largely confirms our results for `text-davinci-003`. Two outliers are the matrix shapes task and the AQuA dataset Ling et al. (2017). For both, Llama-2 exhibits very bad performance across all decoding and prompting strategies, suggesting that the model is likely unable to perform these tasks at all. We attribute this to the difference in model size when compared to OpenAI. `text-davinci-003` has 175 billion parameters, while the Llama-2 variant only has 13 billion parameters. As shown by Kojima et al. (2022), model size directly relates to an increase in general reasoning capabilities.

**Decoders** Combining simple sketches with sketch-aware decoding, we observe even stronger performance gains of up to 10% points, e.g., for BEAMVAR compared to sequential prompting with ARGMAX or BEAM on the question answering datasets AQuA (Ling et al., 2017) and StrategyQA (Geva et al., 2021) with `text-davinci-003`. We observe VAR to perform particularly well on tasks that rely on interleaved reasoning while BEAMVAR is more effective in other settings. For Llama-2, we observe analogous effects, e.g., BEAMVAR improves performance on Date Understanding and GSM8K by almost 7% points, compared to non-templated $CoT$ and simple ARGMAX.

For `text-davinci-003`, we also observe notable performance gains of up to 6% points, when using our sketch-aware decoders in combination with the established *Zero-Shot CoT* (Kojima et al., 2022) prompting scheme (cf. Table 1). This is because *Zero-Shot CoT* already is a two-part prompting scheme, which naturally benefits from our sketch-aware decoders, letting them optimize over the reasoning process (first variable) and final answer (second variable) jointly.

## 4.2 NOVEL APPLICATIONS ENABLED BY PROMPT SKETCHING

In addition to reasoning performance, sketching also enables novel applications, for which non-templated sequential inference either fails completely or is much less effective and reliable. We highlight multiple scenarios here (causal reordering, sudoku, interactive environments) and expand on them in App. B (JSON generation and graph traversal).

$<S_1>$ `[IS_NEEDED1]`
to answer $<Q>$
However, $<S_2>$ `[IS_NEEDED2].`
Therefore, `[CONCLUSION]`

Figure 4: Information Essentiality prompt with forward references (details in App. F).

Table 3: Sketch-aware decoding enables sudoku solving and more effective graph and world traversal.

| | Sequential Prompting | Prompt Sketching (ours) | | |
| --- | --- | --- | --- | --- |
| | ARGMAX | ARGMAX | BEAMVAR | VAR |
| *Sudoku (3x3)* | | | | |
| text-davinci-003 | **98/100** (reordered) | 15/100 | **66/100** | **62/100** |
| text-curie-001 | 9/100 (reordered) | 5/100 | **38/100** | **33/100** |
| *Dungeon Escape* | | | | |
| text-davinci-003 | - | 93/100 ($\sim$4.14 steps) | **100/100 ($\sim$2.75 steps)** | 96/100 ($\sim$3.42 steps) |
| text-curie-001 | - | 38/100 ($\sim$4.0 steps) | **76/100 ($\sim$2.94 steps)** | 46/100 ($\sim$2.30 steps) |
| *TextWorld* | | | | |
| text-davinci-003 | - | 24/32 ($\sim$11.21 steps) | **27/32 ($\sim$9.37 steps)** | 24/32 ($\sim$9.56 steps) |

**Causal Reordering** First, we investigate forward referencing abilities with our sketch-aware decoders. More specifically, we examine whether sketch-aware decoders enable the model to anticipate future information to some degree. For this, we adapt the existing Information Essentiality dataset (Srivastava et al., 2022), by reordering it according to the template shown in Figure 4. The model has to determine the essentiality of two statements <S$_1$> and <S$_2$>, with respect to a given question <Q>. However, in our reordered prompt, the result variable IS_NEEDED1 is decoded before the <Q> is shown. For this custom task (cf. Table 1), we indeed observe that ARGMAX is incapable of producing any meaningful results (0.01 accuracy), whereas, BEAMVAR and VAR achieve an improved accuracy of 0.25 and 0.06 respectively, by exploring a wider hypotheses space.

**Sudoku** We further examine the capabilities of a model to solve simple $3 \times 3$ sudoku-like puzzles: the LLM is tasked to complete a partial grid with unique numbers in $1 - 9$. Similar to before, this task requires forward referencing to effectively choose the correct numbers. As shown in Table 3, out of 100 puzzles with $1 - 6$ empty spots, sequential ARGMAX decoding is only capable of solving 15. This is expected, as greedy decoding does not allow to anticipate any future information (i.e. fixed numbers), before choosing earlier ones. In contrast, BEAMVAR and VAR solve $66/100$ and $62/100$ puzzles respectively, demonstrating again that they explore a wider hypotheses space. A potential alternative is to re-order the template, which allows text-davinci-003 to achieve perfect accuracy with ARGMAX, although re-ordering is not always an option with more complex multi-step puzzles.

**Interactive Environments** Sketch-aware decoders can take the effect of template-induced continuations into account. If we choose these continuations dynamically based on previous model output, we can effectively leverage them to explore interactive environments (Driess et al., 2023). For this, we experiment with LLM-guided graph and world traversal, where an LLM agent traverses a world, starting out in a randomly chosen room, with the goal of finding the exit or completing some other task. For this, we run with random graphs (*Dungeon Escape*), as well as generated *TextWorld* (Côté et al., 2019) environments. For further details on the setup, we refer to App. B.2. As shown in Table 3, in *Dungeon Escape*, ARGMAX mostly finds the exit, but often requires a lot more steps ($\sim 4$) than VAR ($\sim 2.8$) and BEAMVAR ($\sim 2 - 3$) for both OpenAI models, with the smaller text-curie-001 model being less effective overall. In *TextWorld*, wee generally see similar trends, though lower success rates.

### 4.3 DISCUSSION

Our evaluation shows that sketching and, by extension, template-guided LLM inference in general, can significantly improve model reasoning capabilities. Here, we briefly discuss limitations and other considerations relating to design, compuational, and applicability aspects.

**Sketch Design and Iteration** While still sensitive to wording, prompt sketching does offer more control over exact model behavior, thereby addressing some of the difficulties of traditional prompt design (Reynolds and McDonell, 2021; Arora et al., 2023; Zhao et al., 2021). However, sketching is also not a silver bullet: Most importantly, we find that an effective sketch must not be too restrictive to not impair model performance. Still, as substantiated by our results, even simple sketches can already be effective at improving reasoning capabilities. Lastly, much like non-templated prompts, sketches still require iterative development and tuning to achieve optimal performance on a given task. More importantly, however, they offer benefits such as improved control, a guaranteed output format, and reduced free-text formatting instructions, otherwise needed.

**Applicability**   While sketch design still requires some effort, we find that many tasks in our evaluation can be solved with a small set of generic sketches. For instance, we find that a sketched form of chain-of-thought (Wei et al., 2022a) (see Figure 3) is already effective for a wide range of tasks, including arithmetic reasoning and general question answering. For direct adoption, we also publish the sketches used in our evaluation, which can be adapted or used as-is by practitioners.

**Computational Overhead Of Sketch-Aware Decoding**   Sketch-aware decoders naturally incur computational overhead compared to simple, greedy search. While BEAMVAR requires as much compute as regular beam search, VAR requires an additional factor of beam width $n$ more hypotheses to be tracked in parallel. Similar to regular beam search, this is a well-known trade-off: branching decoders are more expensive but still widely used, especially when improved performance and diversity are important.

## 5   RELATED WORK

**Prompting**   Recent works have proposed a variety of different prompting techniques including chain-of-thought prompting (Wei et al., 2022a;b), interactive question answering (Yao et al., 2022b), self-consistency (Wang et al., 2022a), and ThinkSum (Ozturkler et al., 2022). These prompt programming techniques (Reynolds and McDonell, 2021; Zhou et al., 2022), aim to leverage the general reasoning abilities of LLMs to solve diverse tasks. To enable the efficient implementation of such complex prompting techniques, LM-focused programming systems have recently been introduced: PromptChainer (Wu et al., 2022), PromptSource (Bach et al., 2022), and LMQL (Beurer-Kellner et al., 2023) provide development environments for LM interaction. We build on LMQL, as it supports variable constraints and control flow within prompts, enabling the efficient representation of sketches. Finally, language model cascades (Dohan et al., 2022) view LM querying as probabilistic programming over multiple variables, thus implicitly assuming a sketching setting and opening up interesting perspectives for more advanced decoders in the future. In contrast to prompt sketching, however, existing works compose multiple LLM calls in a disconnected manner, and, crucially, do not consider the overall likelihood of the resulting sequence.

**Language Model Decoding**   Most decoding techniques either aim to approximately recover the maximum a posteriori solution under the model distribution or sample from it. Beyond direct sampling from the model distribution, Nucleus Sampling (Holtzman et al., 2020) clips away the tail of the distribution and Locally Typical Sampling (Meister et al., 2022) considers a subset that yields uniform information density sequences. While ARGMAX can be seen as a best-first search of the decoding tree with a maximum width of 1, Beam Search can be seen as a width-first search with a width constrained to $k$ (often 5) trajectories. Best First Beam Search (Meister et al., 2020) combines the two ideas, always exploring the sequence with the largest score while maintaining the width limit, to increase efficiency. Best-k Search (Xu et al., 2022a) drops the width restriction and always explores the $k$ highest scoring sequences. Lattice decoding (Xu et al., 2022b) allows for the recombination of similar trajectories, leading to more diverse solutions. Diverse Beam Search (Vijayakumar et al., 2016) includes a diversity objective in scoring beams. To improve performance on constraint decoding problems, Grid Beam Search (Hokamp and Liu, 2017) creates separate beams for sequences satisfying a different number of constraints. Post and Vilar (2018) propose Dynamic Beam Allocation to instead partition a fixed beam width into pools depending on the number of satisfied constraints, with Hu et al. (2019) introducing a vectorized implementation.

## 6   CONCLUSION

We presented prompt sketching, a novel framework for template-guided LLM inference that phrases templated generation as a segmented sequence decoding problem. This perspective unlocks novel sketch-aware decoding procedures that optimize for overall template likelihood and not just sequentially generate text. Our experiments show that sketching outperforms naive templating as well as sequential prompting like chain-of-thought on 7 out of 8 hard LLM reasoning tasks, improving task accuracy by up to $10\%$ points. Looking forward, we also show how sketching enables novel applications such as reliable output formatting, forward references in reasoning, and LLM-guided graph traversal, inspiring future work in this direction.

## 7    REPRODUCIBILITY

We publish our code, prompts, and detailed instructions on how to reproduce our results at ANONYMIZED, providing an anonymized version to the reviewers in the supplementary material, including our implementation of `dclib` (also see App. E). Additionally, we provide detailed descriptions of all prompts in App. F.

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

## A  DECODING ALGORITHMS

**Variable-Level Beam Search VAR**  The pseudo-code implementation of VAR is given in Algorithm 2. The function $expand_{det}$, expands a given sequence by a deterministic chunk if the next chunk in the prompt template is not a variable. The function $expand_{sample}$, expands a given sequence by sampling $n$ different continuations for the next variable value in the prompt template. Lastly, $top_n$ selects the $n$ best sequences from a given set of sequences, according to the the length normalized beam search score as discussed in Section 2. In practice, an additional early stopping criterion on $done$ is employed.

---

**Algorithm 1** Variable-Level Beam Search (VAR)

---

**Require:** Input $n$: number of beams, $\mathcal{V}$ set of variables
**Ensure:** set of $n$ VAR-best hypotheses $done$
1: $h \leftarrow \{\textit{<bos>}\}, h' \leftarrow \{\}$
2: **for** $v_i \in \mathcal{V}$ **do**
3:    $h \leftarrow expand_{det}(h)$
4:    **for** $s \in h$ **do**
5:       $h' \leftarrow h' + expand_{sample}(s, n)$
6:    **end for**
7:    $h \leftarrow top_n(h')$
8:    $done, h \leftarrow separate\_done(h)$
9: **end for**

---

**Variable-Grid Beam Search VAR**  The simplified pseudo-code implementation of BEAMVAR is given in Algorithm 2. The function $expand_{det}$, expands a given sequence by a deterministic chunk if the next chunk in the prompt template is not a variable. The function $expand_{top}$, expands a given sequence by the *top-n* token continuations according to the model distribution. $post\_vilar$ determines the dynamic beam size per group according to Post and Vilar (2018), where groups are defined by the currently decoded variable and or deterministic chunk. Lastly, $top_n$ selects the $n$ best sequences from a given set of sequences, according to the the length normalized beam search score as discussed in Section 2. In practice, an additional early stopping criterion on $done$ is employed.

---

**Algorithm 2** Variable-Grid Beam Search (BEAMVAR)

---

**Require:** Input $n$: number of beams, $N$: maximum length, $\mathcal{V}$ set of variables
**Ensure:** set of $n$ BEAMVAR-best hypotheses $done$
1: $h \leftarrow \{\textit{<bos>}\}, h' \leftarrow \{\}$
2: **for** $i \in \{1, \ldots, N\}$ **do**
3:    $h \leftarrow expand_{det}(h)$
4:    $h' \leftarrow \{\}$
5:    {*Expand each hypothesis in $h$ by its top-n continuations*}
6:    **for** $s \in h$ **do**
7:       $h' \leftarrow h' + expand_{top}(s, n)$
8:    **end for**
9:    $h \leftarrow \{\}$
10:    {*Determine dynamic beam size per group according to Post and Vilar (2018)*}
11:    $\{n_0 \ldots n_{|\mathcal{V}|}\} \leftarrow post\_vilar(h')$
12:    **for** $v_i \in \mathcal{V}$ **do**
13:       $h \leftarrow h + top_{n_{v_i}}(h')$
14:    **end for**
15:    {*Filter out completed sequences*}
16:    $done, h \leftarrow separate\_done(h)$
17: **end for**

---

# B APPLICATIONS

## B.1 SKETCHING FOR OUTPUT FORMATTING

One direct application of sketching, is to generate schema-conform JSON objects with LLMs, given only free text as input, as illustrated in Figure 5. Using sketching, we can guarantee the output format 10/10 times with different examples similar to the one shown in Figure 5. This works for both, `text-davinci-003` and `text-curie-001`, regardless of the model (size) used. Without sketching, `text-davinci-003` requires detailed descriptions of the output format in the prompt to produce valid JSON at a similar rate. At the same time it may still fail stochastically (no guarantees), and the smaller `text-curie-001` is not be able to produce valid JSON at all when just instructed to do so. Further, including detailed data format instructions in non-templated queries in this way, causes a high average inference cost of 179.5 tokens per sample, reduced to 28.7 when using sketching, an 84% reduction in inference cost.

Alex Kim is a software architect at Intel, designing and implementing complex systems for the company's processors. He graduated from the University of California, Los Angeles with a degree in Computer Science and enjoys playing video games and practicing photography.

```
As JSON:
{
  "name": "[VALUE] Alex Kim",
  "job": "[VALUE] software architect",
  "role": "[VALUE] systems engineer",
  "education": {
    "university": "[VALUE] University of
                   California, Los
                   Angeles",
    "degree": "[VALUE] Computer Science",
  },
  "interests": "[VALUE] video games,
                photography",
}
```

Figure 5: Sketched JSON parsing. Only highlighted text is completed by the model.

## B.2 INTERACTIVE ENVIRONMENTS

As part of our evaluation, we also consider the use of sketching and our sketch-aware decoders in interactive environments.

**Dungeon Escape** For our Dungeon Escape experiment, we generate 100 random dungeons with $8 - 10$ rooms each, where the average shortest exit route is 2.3 steps away. At each node, the model is asked for the next room/node to traverse to. We rely on the following interactive sketch program with corresponding constraints on sketch variable ACTION:

```
node = <initialized to start node>
steps = 0
max_steps = 10

while rooms[node] != 'Exit':
    name = rooms[node]
    neighbours = hallways[node]
    "System: You are in room {node} '{name}'. "
    "You can go to {neighbours}. "
    "Where do you want to go?\n"
    "You:[ACTION]\n"
    next_node = int(ACTION.strip())
    if next_node not in neighbours:
        "System: {next_node} is not a valid neighboring
         room of '{name}'. Valid rooms are {neighbours}.\n"
    else:
        node = next_node
    steps += 1

    if steps > max_steps:
        "System: You have taken too many steps. You lose.\n"
        return "failure"
return "success"
```

**Constraints:** ACTION **in** ["0", "1", "2", "3", "4", "5", "6", "7", "8", "9"]

Depending on the graph that is being explored, this results in a runtime instantiation of a prompt sketch as shown in Figure 6. The sketch-aware decoder is then used to generate the next action to take, given the current state of the environment. The generated action is then executed in the environment and the process is repeated until the agent reaches the exit or the maximum number of steps is exceeded. Depending on the decoding algorithm, the agent can be made to explore the

environment in different ways. For example, ARGMAX will result in a greedy, depth-first search, while VAR and BEAMVAR result in a breadth-first search.

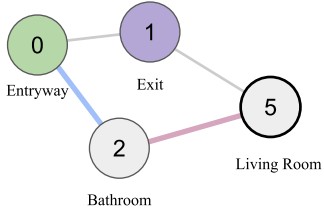

Figure 6: Exploring a graph using prompt sketching.

**TextWorld** For our *TextWorld* experiments, we rely on the following command, to generate custom environments of different sizes and difficulty levels:

```
tw-make custom --world-size 5 --nb-objects 10 --quest-length {i} --seed 4321 --output
eval_games/game-{i}.z8
```

We choose i ∈ [2,10] and generate 4 worlds per size. We decode a sketch that uses an LLM to decode the next *TextWorld* action to perform. We report the detailed results about the number of steps and the number of games solved in Figure 7. As shown, especially towards larger game size (more actions required), BEAMVAR and VAR solve more games and require less steps to do so, compared to ARGMAX. This is in line with our results on the *Dungeon Escape* environment. ARGMAX performs a greedy, single, depth-first traversal with in-context backtracking, while VAR and BEAMVAR perform a breadth-first search, allowing them to complete quests quicker and more reliably.

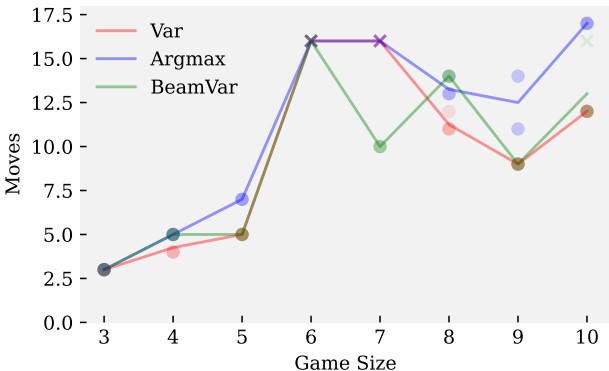

Figure 7: Solving *TextWorld* with ARGMAX, BEAMVAR and VAR. Circles mark solved games, while crosses mark unsolved games which are counted as games with step limit exceeded (16)

Table 4: Few-Shot Task Accuracy with *Answer-Only*, *Chain-Of-Thought* and *Prompt Sketching* (ours) using ARGMAX, BEAM, BEAMVAR (ours) and VAR (ours) decoding. The best results are highlighted in bold.

| Task | Two-Shot | | | | Zero-Shot | |
|------|----------|--|--|--|-----------|--|
| | Sequential Decoding | | | | **Sketch-Aware** | |
| Task | *Answer-Only* | *CoT* | *Prompt Sketching (ours)* | | | |
| | ARGMAX | ARGMAX | ARGMAX | VAR | **BEAMVAR** | VAR |
| *Question Answering* | | | | | | |
| AQuA (Ling et al., 2017) | 0.29 | 0.45 | 0.46 | 0.44 | **0.47** | 0.35 |
| StrategyQA (Geva et al., 2021) | 0.67 | 0.74 | **0.78** | **0.78** | 0.77 | 0.66 |
| *Interleaved Reasoning* | | | | | | |
| Tracking Shuffled Objects[+] | 0.1 | 0.46 | 0.57 | 0.57 | 0.62 | **0.66** |
| Matrix Shapes[+] | 0.67 | 0.76 | 0.81 | 0.77 | 0.79 | **0.85** |

[+] Tasks extracted from the BIG benchmark suite (Srivastava et al., 2022).
[*] Specifically adapted for our evaluation of forward referencing prompts.

## C  ADDITIONAL RESULTS

### C.1  FEW-SHOT PROMPTING

In addition to evaluating zero-shot performance of sketching, we also evaluate a two-shot setting (two demonstrations) for selected datasets (AQuA, StrategyQA, Tracking Shuffled Objects and Matrix Shapes) and report the results in Table 4. We rely on few-shot samples exhibiting the same reasoning structure as enforced by the respective sketches. For comparison, we also include the best zero-shot result from the paper.

While we observe a slight increase in performance for the question-answering tasks, performance for the interleaved reasoning tasks is surprisingly decreased in the few-shot setting. In all considered settings, sketching outperforms CoT. In fact, zero-shot sketching with the best decoder even outperforms few-shot CoT with argmax decoding in all settings. Upon manual inspection, we observe that the LLM generally follows the demonstrated reasoning process. However, for Tracking Shuffled Objects, the added demonstrations seem to impair task performance, possibly because the model is confused by the unrelated extra information. Overall, the results of this ablation study suggest that zero-shot sketching with the right decoder may be able to replace few-shot demonstrations by enforcing a given reasoning structure via intermediate instructions and task decomposition. At the same time, note that sketching is much more cost-effective, as few-shot samples increase overall sequence lengths, and thus API cost when using OpenAI models or computational cost (scaling quadratically with sequence length) when using local models.

### C.2  PROMPT SKETCHING WITH SMALLER OPENAI MODELS

We also examine the use of smaller OpenAI models. However, the strong zero-shot performance we rely on has previously (for CoT) only been observed in models of sufficient size (150B+ parameters (Kojima et al., 2022)). Nonetheless, we also run our evaluation to the smaller InstructGPT (Ouyang et al., 2022) model `text-curie-001` (1 level below `text-davinci-00x`, 6.7 billion parameters). For comparison, we also include the best results for `text-davinci-003`, as reported in the paper in Table 5.

Overall, we observe almost no reasoning capabilities, with scores close to random guessing for multiple-choice tasks, and close to zero for open-ended questions like Matrix Shapes. As noted above, this is in line with previous results (Kojima et al., 2022). However, as our main evaluation demonstrates, the slightly larger and more recent *Llama-2 Chat 13B* Touvron et al. (2023) (13 billion parameters), does clearly benefit from sketching.

Table 5: Task Accuracy with `text-curie-001` with *Answer-Only*, *Chain-Of-Thought* and *Prompt Sketching* (ours) using ARGMAX, BEAM, BEAMVAR (ours) and VAR (ours) decoding, compared to the results with `text-davinci-003`. The best results are highlighted in bold.

| Task | text-curie-001 | | | | text-davinci-003 | |
|------|----------------|---|---|---|------------------|---|
| | Sequential Decoding | | | | Sketch-Aware | |
| Task | *Answer-Only* | *CoT* | *Prompt Sketching (ours)* | | | |
| | ARGMAX | ARGMAX | ARGMAX | VAR | BEAMVAR | VAR |
| *Question Answering* | | | | | | |
| AQuA (Ling et al., 2017) | 0.16 | 0.24 | 0.27 | 0.17 | **0.47** | 0.35 |
| StrategyQA (Geva et al., 2021) | 0.46 | 0.53 | 0.58 | 0.52 | **0.77** | 0.66 |
| *Interleaved Reasoning* | | | | | | |
| Tracking Shuffled Objects[+] | 0.18 | 0.19 | 0.22 | 0.24 | 0.62 | **0.66** |
| Matrix Shapes[+] | 0.04 | 0.07 | 0.01 | 0.0 | 0.79 | **0.85** |

[+] Tasks extracted from the BIG benchmark suite (Srivastava et al., 2022).
[*] Specifically adapted for our evaluation of forward referencing prompts.

### C.3 CONFIDENCE BOUNDS AND SCALING OF OPENAI-SPECIFIC RESULTS

To check for significance of our smaller scale OpenAI-specific experiments, we additionally examine the corresponding confidence bounds. For this, we report all main OpenAI results with a Clopper-Pearson 95% two-sided confidence interval in Table 6.

Lastly, we scale our best OpenAI results for the *Matrix Shapes* task Srivastava et al. (2022), by evaluating with 1000 instead of 100 samples, sampled uniformly from the original dataset. Doing so, we can confirm our main result in Table 7, i.e. that prompt sketching and interleaved reasoning specifically are effective at improving LLM reasoning performance on this task. Due to cost considerations, we cannot scale all OpenAI experiments like this, but expect similar results, similar to the trends we observe in our large scale experiments with *Llama-2 Chat 13B* ( Touvron et al. (2023) (see Section 4).

Table 6: Task accuracy of `text-davinci-003` with Clopper-Pearson 95% two-sided confidence intervals Clopper and Pearson (1934).

| Task | Sequential Decoding | | | | | Sketch-Aware Decoding (ours) | | | |
|------|---------------------|---|---|---|---|------------------------------|---|---|---|
| | *Answer-Only* | *Chain-Of-Thought Prompting* | | | | *Prompt Sketching (ours)* | | | |
| | ARGMAX | ARGMAX | BEAM | BEAMVAR | VAR | ARGMAX | BEAMVAR | VAR | BEAM |
| *Logical Reasoning* | | | | | | | | | |
| Date Understanding[*] | 0.48 [0.38,0.58] | 0.70 [0.6,0.79] | **0.75** [0.65,0.83] | **0.75** [0.65,0.83] | 0.69 [0.59,0.78] | 0.72 [0.62,0.81] | 0.73 [0.63,0.81] | 0.66 [0.56,0.75] | - |
| Information Essentiality[*+] | - | - | - | - | - | 0.01 [0.0,0.05] | **0.25** [0.17,0.35] | 0.06 [0.02,0.13] | 0.15 [0.09,0.24] |
| *Question Answering* | | | | | | | | | |
| AQuA (Ling et al., 2017) | 0.31 [0.22,0.41] | 0.37 [0.28,0.47] | 0.37 [0.28,0.47] | 0.35 [0.26,0.45] | 0.35 [0.26,0.45] | 0.40 [0.3,0.5] | **0.47** [0.37, 0.57] | 0.35 [0.26,0.45] | - |
| StrategyQA (Geva et al., 2021) | 0.68 [0.58,0.77] | 0.71 [0.61,0.8] | 0.72 [0.62,0.81] | 0.67 [0.57,0.76] | 0.67 [0.57,0.76] | 0.69 [0.59,0.78] | **0.77** [0.68,0.85] | 0.66 [0.56,0.75] | - |
| *Arithmetic Reasoning* | | | | | | | | | |
| Multistep Arithmetic[*] | 0.20 [0.13,0.29] | 0.43 [0.33,0.53] | 0.44 [0.34,0.54] | **0.49** [0.39,0.59] | 0.44 [0.34,0.54] | 0.45 [0.35,0.55] | 0.48 [0.38,0.58] | 0.38 [0.28,0.48] | - |
| GSM8K (Cobbe et al., 2021) | 0.08 [0.04,0.15] | 0.56 [0.46,0.66] | 0.58 [0.48,0.68] | **0.64** [0.54,0.73] | 0.57 [0.48,0.68] | 0.57 [0.47,0.67] | 0.53 [0.43,0.63] | 0.59 [0.49,0.69] | - |
| *Interleaved Reasoning* | | | | | | | | | |
| Shuffled Objects[*] | 0.19 [0.12,0.28] | 0.62 [0.52,0.72] | 0.47 [0.37,0.57] | 0.52 [0.42,0.62] | 0.62 [0.52,0.72] | 0.64 [0.54,0.73] | 0.62 [0.52,0.72] | **0.66** [0.56,0.75] | - |
| Matrix Shapes[*] | 0.61 [0.51,0.71] | 0.77 [0.68,0.85] | 0.77 [0.61,0.8] | 0.71 [0.66,0.84] | 0.76 [0.66,0.84] | 0.81 [0.72,0.88] | 0.79 [0.7,0.87] | **0.85** [0.76,0.91] | - |

[+] Tasks extracted from the BIG benchmark suite (Srivastava et al., 2022).
[*] Specifically adapted for our evaluation of forward referencing prompts.

Table 7: Task Accuracy when evaluating with 1000 samples from the original dataset.

| Task | Sequential Decoding | | | | | Sketch-Aware Decoding (ours) | | | |
|------|---------------------|---|---|---|---|------------------------------|---|---|---|
| | *Answer-Only* | *Chain-Of-Thought Prompting* | | | | *Prompt Sketching (ours)* | | | |
| | ARGMAX | ARGMAX | BEAM | BEAMVAR | VAR | ARGMAX | BEAMVAR | VAR | BEAM |
| *Interleaved Reasoning* | | | | | | | | | |
| Matrix Shapes[+] | 0.572 [0.54,0.6] | 0.779 [0.75,0.8] | - | - | - | 0.814 [0.79,0.84] | - | **0.817** [0.79,0.84] | - |

[+] Tasks extracted from the BIG benchmark suite (Srivastava et al., 2022).

### C.4 CONFIDENCE BOUNDS FOR LLAMA-2

As for the OpenAI models we report the confidence bounds for the Llama-2 Chat model (13 billion parameters) in Table 8.

Table 8: Task accuracy of Llama-2 Chat (13B) with Clopper-Pearson 95% two-sided confidence intervals Clopper and Pearson (1934). We either use 1000 uniformly drawn samples from the dataset or the whole dataset.

| Task | Sequential Decoding | | | | | Sketch-Aware Decoding (ours) | | | |
|---|---|---|---|---|---|---|---|---|---|
| | Answer-Only | Chain-Of-Thought Prompting | | | | Prompt Sketching (ours) | | | |
| | ARGMAX | ARGMAX | BEAM | BEAMVAR | VAR | ARGMAX | BEAMVAR | VAR | BEAM |
| *Logical Reasoning* | | | | | | | | | |
| Date Understanding[+] | 0.496 [0.444, 0.548] | 0.591 [0.539, 0.641] | 0.599 [0.547, 0.649] | 0.613 [0.561, 0.662] | 0.580 [0.528, 0.631] | 0.634 [0.583, 0.683] | **0.656** [0.605, 0.704] | 0.642 [0.591, 0.691] | 0.591 [0.539, 0.641] |
| Information Essentiality[++] | - | - | - | - | - | 0.088 [0.033, 0.182] | **0.132** [0.062, 0.236] | **0.132** [0.062, 0.236] | **0.132** [0.062, 0.236] |
| *Question Answering* | | | | | | | | | |
| AQuA (Ling et al., 2017) | 0.231 [0.180, 0.288] | 0.291 [0.235, 0.351] | **0.311** [0.254, 0.372] | 0.275 [0.221, 0.335] | 0.271 [0.217, 0.330] | 0.239 [0.188, 0.297] | 0.255 [0.202, 0.314] | 0.243 [0.191, 0.301] | 0.283 [0.229, 0.344] |
| StrategyQA (Geva et al., 2021) | 0.564 [0.533, 0.595] | 0.555 [0.524, 0.586] | 0.566 [0.535, 0.597] | 0.570 [0.539, 0.601] | 0.568 [0.537, 0.599] | 0.638 [0.607, 0.668] | 0.630 [0.599, 0.660] | 0.640 [0.609, 0.670] | **0.659** [0.629, 0.688] |
| *Arithmetic Reasoning* | | | | | | | | | |
| Multistep Arithmetic[+] | 0.038 [0.027, 0.052] | 0.133 [0.113, 0.156] | 0.120 [0.101, 0.142] | 0.138 [0.117, 0.161] | 0.132 [0.111, 0.154] | 0.126 [0.106, 0.148] | **0.142** [0.121, 0.165] | 0.103 [0.085, 0.124] | 0.125 [0.105, 0.147] |
| GSM8K (Cobbe et al., 2021) | 0.049 [0.036, 0.064] | 0.276 [0.248, 0.305] | 0.296 [0.268, 0.325] | 0.305 [0.277, 0.335] | 0.296 [0.268, 0.325] | 0.320 [0.291, 0.350] | **0.353** [0.323, 0.384] | 0.350 [0.320, 0.380] | **0.353** [0.323, 0.384] |
| *Interleaved Reasoning* | | | | | | | | | |
| Shuffled Objects[+] | 0.197 [0.173, 0.223] | 0.196 [0.172, 0.222] | 0.204 [0.179, 0.230] | 0.198 [0.174, 0.224] | 0.188 [0.164, 0.214] | 0.227 [0.201, 0.254] | 0.210 [0.185, 0.237] | **0.234** [0.208, 0.262] | 0.157 [0.135, 0.181] |
| Matrix Shapes[+] | **0.227** [0.201, 0.254] | 0.068 [0.053, 0.085] | 0.065 [0.051, 0.082] | 0.056 [0.043, 0.072] | 0.191 [0.167, 0.217] | 0.205 [0.180, 0.231] | 0.200 [0.176, 0.226] | 0.204 [0.179, 0.23] | 0.200 [0.176, 0.226] |

[+] Tasks extracted from the BIG benchmark suite (Srivastava et al., 2022).
[+] Specifically adapted for our evaluation of forward referencing prompts.

Table 9: Self-Consistency Wang et al. (2022b) prompting compared to simple ARGMAX *chain-of-thought* prompting (`text-davinci-003`, 100 samples as in Table 1). Results that are greyed out are not comparable to the respective sketching results, due to different levels of computation.

| | *Chain-Of-Thought* | | | **Chain-Of-Thought/Sketching (ours)** | |
|---|---|---|---|---|---|
| Task | ARGMAX | SELFCONSIST ($n=2$) | SELFCONSIST ($n=4$) | Sketch-Aware | |
| *Logical Reasoning* | | | | | |
| Date Understanding[+] | 0.70 | 0.64 | 0.72 | **0.75** | (*CoT*+BEAMVAR) |
| *Question Answering* | | | | | |
| AQuA (Ling et al., 2017) | 0.37 | 0.33 | 0.32 | **0.47** | (*Sketching*+BEAMVAR) |
| StrategyQA (Geva et al., 2021) | 0.71 | 0.64 | 0.70 | **0.77** | (*Sketching*+BEAMVAR) |
| *Arithmetic Reasoning* | | | | | |
| Multistep Arithmetic[+] | 0.43 | 0.40 | 0.47 | **0.49** | (*CoT*+BEAMVAR) |
| GSM8K (Cobbe et al., 2021) | 0.56 | 0.56 | **0.66** | **0.64** | (*CoT*+BEAMVAR) |
| *Interleaved Reasoning* | | | | | |
| Tracking Shuffled Objects[+] | 0.62 | 0.39 | 0.43 | **0.66** | (*Sketching*+VAR) |
| Matrix Shapes[+] | 0.77 | 0.66 | 0.66 | **0.85** | (*Sketching*+VAR) |

[+] Tasks extracted from the BIG benchmark suite (Srivastava et al., 2022).
[*] Specifically adapted for our evaluation of forward referencing prompts.

## D  SELF-CONSISTENCY BASELINE

We also considered self-consistency Wang et al. (2022b) ($SC$) as a possible baseline in our experiments. Based on our decoder experiments, we ran self-consistency with chain-of-thought and $n=2$ and $n=4$ consistency samples. Computationally, this is comparable to BEAMVAR $n=2$ and VAR $n=2$ respectively, as in our main evaluation. We report the results in Table 9. We find $SC$ mostly does not even outperform a simple ARGMAX *chain-of-thought* baseline, except with $n=4$ on the GSM8K dataset. There, it achieves 0.66, outcompeting even BEAMVAR with $n=2$ (0.64). However, we note that BEAMVAR with $n=2$, is also half as computationally expensive as $SC$ with $n=4$, while achieving comparable performance.

Increasing the number of self-consistency samples to a higher $n$ would be possible but is not a useful comparison, as it would increase the cost of $SC$ significantly, over our sketch-aware decoders (see Section 4.3). Lastly, we note that self-consistency and sketch-aware decoding are orthogonal, and could be combined to further improve performance.

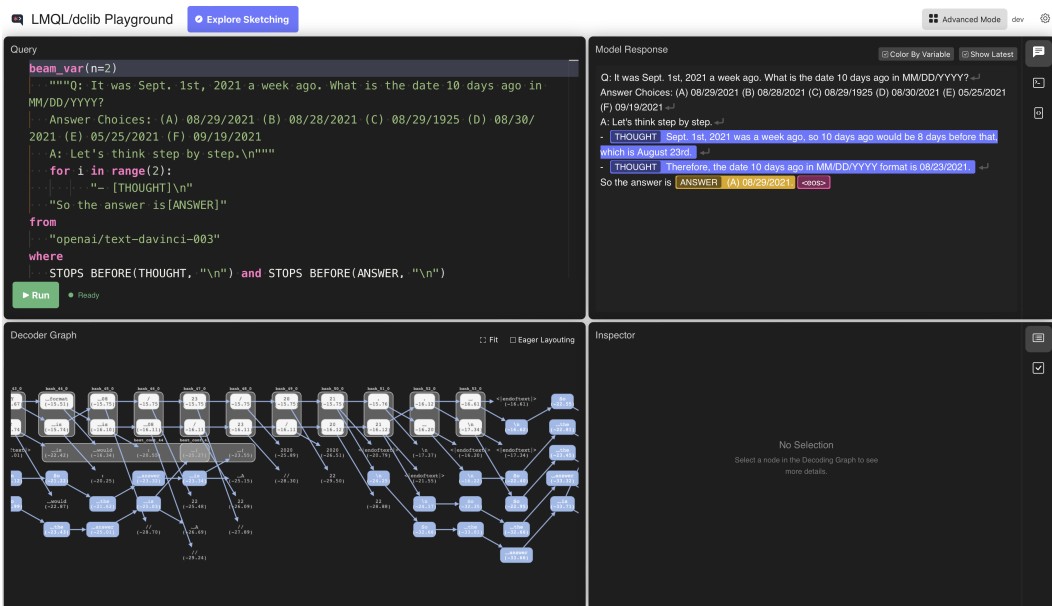

Figure 8: The adapted LMQL playground interface extended with `dclib` support allows users to visualize the underlying decoding trees during sketch decoding.

# E   USING `dclib` AND SKETCH-AWARE DECODERS

In the supplementary material, we provide the Python library `dclib`, that contains implementations for all compared (sketch-aware) decoding algorithms.

To install and use `dclib`, please consult the `README.md` file in the supplementary material.

Once installed, you can use an adapted version of the `lmql playground` as shown in Figure 8 to interactively play with the different decoding algorithms and their parameters.

## F   FULL PROMPTS

Here we list full exemplary prompts, per task and prompting method as used in our evaluation. We rely on the same notation of sketches as in the main body of the paper. For control-flow (e.g. loops and conditions) and constraints, we rely on the semantics of the LMQL query language for LMs, and refer to Beurer-Kellner et al. (2023) for a more thorough introduction.

| Task | Prompting Method | Query and Response |
|---|---|---|
| date_understanding@ao | *Answer-Only* | **Query**

```"""Q: It was Sept. 1st, 2021 a week ago. What is the date 10 days ago in MM/DD/YYYY? Answer Choices: (A) 08/29/2021 (B) 08/28/2021 (C) 08/29/1925 (D) 08/30/2021 (E) 05/25/2021 (F) 09/19/2021 Among A through F, the answer is[answer]"""```

**Constraints**

```answer in [" A", " B", " C", " D", " E", " F"]```

**Model Response** Q: It was Sept. 1st, 2021 a week ago. What is the date 10 days ago in MM/DD/YYYY? Answer Choices: (A) 08/29/2021 (B) 08/28/2021 (C) 08/29/1925 (D) 08/30/2021 (E) 05/25/2021 (F) 09/19/2021 Among A through F, the answer is B ✗ |
| date_understanding@cot | *Chain-Of-Thought* | **Query**

```"""Q: It was Sept. 1st, 2021 a week ago. What is the date 10 days ago in MM/DD/YYYY? Answer Choices: (A) 08/29/2021 (B) 08/28/2021 (C) 08/29/1925 (D) 08/30/2021 (E) 05/25/2021 (F) 09/19/2021 A: Let's think step-by-step. [COT] Therefore, among A through F, the answer is[answer]"""```

**Model Response** Q: It was Sept. 1st, 2021 a week ago. What is the date 10 days ago in MM/DD/YYYY? Answer Choices: (A) 08/29/2021 (B) 08/28/2021 (C) 08/29/1925 (D) 08/30/2021 (E) 05/25/2021 (F) 09/19/2021 A: Let's think step-by-step.
Sept. 1st, 2021 was a week ago, so 10 days ago would be 8 days before that, which would be August 23rd, 2021.
Therefore, the answer is (A) 08/23/2021. Therefore, among A through F, the answer is A. ✓ |

| Task | Prompting Method | Query and Response |
|---|---|---|
| date_under-standing@multivar2 | *Multi-Variable* | **Query**

`"""Q: It was Sept. 1st, 2021 a week ago. What is the date 10 days ago in MM/DD/YYYY?`
`Answer Choices: (A) 08/29/2021 (B) 08/28/2021 (C) 08/29/1925 (D) 08/30/2021 (E) 05/25/2021 (F) 09/19/2021`
`A: Let's think step by step."""`
`for i in range(5):`
`"\\n-[THOUGHT]"`
`if not THOUGHT.endswith("\\n"): "\\n"`
`if "Therefore" in THOUGHT: break`
`if "According" in THOUGHT: break`
`" Overall this means,[CONCLUSION] Therefore, among A through F, the answer is[answer]"`

**Constraints**

`STOPS_AT(THOUGHT, "\\n") and STOPS_AT(THOUGHT, ".") and STOPS_AT(CONCLUSION, "\\n") and STOPS_AT(CONCLUSION, ".")`

**Model Response** Q: It was Sept. 1st, 2021 a week ago. What is the date 10 days ago in MM/DD/YYYY? Answer Choices: (A) 08/29/2021 (B) 08/28/2021 (C) 08/29/1925 (D) 08/30/2021 (E) 05/25/2021 (F) 09/19/2021 A: Let's think step by step. - Sept. - 1st - 2021 - a week ago - 10 days ago Overall this means, 10 days before Sept. Therefore, among A through F, the answer is B: 08/28/2021. ✗ |

| Task | Prompting Method | Query and Response |
|---|---|---|
| Information Essentiality | *Forward Referencing* | **Query**

`"Cole is on the varsity basketball team. Therefore, the statement that Cole is under 7 feet tall[var_1] to answer the question: Is Cole over 6 feet tall? However, the statement that Every player on Cole's varsity basketball team is above 6'1''[var_2]. Therefore [answer]"`

**Constraints**

`var_1 in [" is  sufficient", " is insufficient", " is not needed"] and var_2 in [" would also be sufficient", " would alone be insufficient", " in addition is sufficient", " is sufficient" , " is insufficient", " is still insufficient", " is also not needed" ] and answer in ["the first statement alone is sufficient while the second statement alone is insufficient", "the second statement alone is sufficient while the first statement alone is insufficient", "either the first or the second statement is sufficient", "the first and second statement taken together are sufficient", "the question can not be answered with the provided statements", "the question can be answered without either statement"]`

**Model Response** Cole is on the varsity basketball team. Therefore, the statement that Cole is under 7 feet tall is not needed to answer the question: Is Cole over 6 feet tall? However, the statement that Every player on Cole's varsity basketball team is above 6'1" is also not needed. Therefore the question can be answered without either statement ✗ |

| Task | Prompting Method | Query and Response |
|---|---|---|
| AQUA@ao | *Answer-Only* | **Query**

```"""Q: What is the greatest number of identical bouquets that can be made out of 28 white and 98 red tulips if no flowers are to be left out? (Two bouquets are identical whenever the number of red tulips in the two bouquets is equal and the number of white tulips in the two bouquets is equal.)\\n"""```
```"Answer Choices: (A) 4 (B) 7 (C) 10 (D) 14 (E) 21\\n"```
```"A: Among A through E, the answer is[answer]"```

**Model Response** Q: What is the greatest number of identical bouquets that can be made out of 28 white and 98 red tulips if no flowers are to be left out? (Two bouquets are identical whenever the number of red tulips in the two bouquets is equal and the number of white tulips in the two bouquets is equal.) Answer Choices: (A) 4 (B) 7 (C) 10 (D) 14 (E) 21 A: Among A through E, the answer is D, 14. ✓ |
| AQUA@cot | *Chain-Of-Thought* | **Query**

```"""Q: What is the greatest number of identical bouquets that can be made out of 28 white and 98 red tulips if no flowers are to be left out? (Two bouquets are identical whenever the number of red tulips in the two bouquets is equal and the number of white tulips in the two bouquets is equal.)\\n"""```
```"Answer Choices: (A) 4 (B) 7 (C) 10 (D) 14 (E) 21\\n"```
```"A: Let's think step by step.[COT] Therefore, among A through E, the answer is[answer]"```

**Constraints**

```STOPS_AT(answer, ".")```

**Model Response** Q: What is the greatest number of identical bouquets that can be made out of 28 white and 98 red tulips if no flowers are to be left out? (Two bouquets are identical whenever the number of red tulips in the two bouquets is equal and the number of white tulips in the two bouquets is equal.) Answer Choices: (A) 4 (B) 7 (C) 10 (D) 14 (E) 21 A: Let's think step by step.
We know that we have 28 white tulips and 98 red tulips. We also know that two bouquets are identical when the number of red tulips and white tulips are equal.
Therefore, we can make the greatest number of identical bouquets by making sure that each bouquet has the same number of red and white tulips.
The greatest number of identical bouquets that can be made out of 28 white and 98 red tulips is 10. (C) Therefore, among A through E, the answer is C. ✗ |

| Task | Prompting Method | Query and Response |
|---|---|---|
| AQUA@multivar | *Multi-Variable* | **Query**

```py
"""Q: What is the greatest number of identical bouquets
 that can be made out of 28 white and 98 red tulips if
no flowers are to be left out? (Two bouquets are
identical whenever the number of red tulips in the two
bouquets is equal and the number of white tulips in the
 two bouquets is equal.)\\n"""
"Answer Choices: (A) 4 (B) 7 (C) 10 (D) 14 (E) 21\\n"
"A: Let's think step by step.\\n"
for i in range(12):
"\\n- [THOUGHT]"
if not THOUGHT.endswith("\\n"): "\\n"
if "answer" in THOUGHT: break
" Overall this means,[CONCLUSION] Therefore, among A
through E, the answer is[answer]"
```

**Constraints**

```py
STOPS_AT(answer, ".") and STOPS_AT(THOUGHT, "\\n") and
STOPS_AT(CONCLUSION, "\\n") and STOPS_AT(CONCLUSION, ".
")
```

**Model Response** Q: What is the greatest number of identical bouquets that can be made out of 28 white and 98 red tulips if no flowers are to be left out? (Two bouquets are identical whenever the number of red tulips in the two bouquets is equal and the number of white tulips in the two bouquets is equal.) Answer Choices: (A) 4 (B) 7 (C) 10 (D) 14 (E) 21 A: Let's think step by step.
- We have 28 white and 98 red tulips.
- We need to make bouquets with the same number of white and red tulips in each bouquet.
- The greatest number of identical bouquets that can be made is 14.
- This is because we can make 7 bouquets with 4 white and 14 red tulips each, and 7 bouquets with 14 white and 4 red tulips each.
- Therefore, the answer is (D) 14. Overall this means, 28 white and 98 red tulips can make 14 identical bouquets. Therefore, among A through E, the answer is D. ✓ |

| Task | Prompting Method | Query and Response |
|------|------------------|--------------------|
| StrategyQA@ao | *Answer-Only* | **Query**

`"""Q: While on a liquid diet, are there some types of`
`soup you cannot eat?\\n"""`
`"A: The answer (yes or no) is[answer]"`

**Constraints**

`answer in [' yes', ' no', ' false', ' true']`

**Model Response** Q: While on a liquid diet, are there some types of soup you cannot eat? A: The answer (yes or no) is yes ✓ |
| StrategyQA@cot | *Chain-Of-Thought* | **Query**

`"""Q: While on a liquid diet, are there some types of`
`soup you cannot eat?\\n"""`
`"A: Let's think step by step.\\n[COT] \\n Therefore,`
`the answer (yes or no) is[answer]"`

**Constraints**

`answer in [' yes', ' no', ' false', ' true']`

**Model Response** Q: While on a liquid diet, are there some types of soup you cannot eat? A: Let's think step by step.
First, what is a liquid diet? A liquid diet is a diet that consists of only liquids, such as water, juices, and soups.
Now, are there some types of soup you cannot eat while on a liquid diet? Generally speaking, no, there are no types of soup that you cannot eat while on a liquid diet. However, it is important to note that some soups may contain ingredients that are not allowed on a liquid diet, such as dairy products, nuts, and certain types of meat. Therefore, it is important to read the ingredients list of any soup you are considering eating while on a liquid diet to make sure it does not contain any of these ingredients. Therefore, the answer (yes or no) is no ✗ |

| Task | Prompting Method | Query and Response |
|------|------------------|--------------------|
| StrategyQA@multivar | *Multi-Variable* | **Query**

```"""Q: While on a liquid diet, are there some types of soup you cannot eat?\\n"""```
```"A: Let's think step by step.\\n"```
```for i in range(5):```
```"\\n-[THOUGHT]"```
```if not THOUGHT.endswith("\\n"): "\\n"```
```if "Therefore" in THOUGHT: break```
```if "According" in THOUGHT: break```
```" Overall this means,[CONCLUSION] Therefore, the answer (yes or no) is[answer]"```

**Constraints**

```STOPS_AT(THOUGHT, "\\n") and STOPS_AT(THOUGHT, ".") and answer in [' yes', ' no'] and```
```STOPS_AT(CONCLUSION, "\\n") and STOPS_AT(CONCLUSION, ".")```

**Model Response** Q: While on a liquid diet, are there some types of soup you cannot eat? A: Let's think step by step.
- First, what is a liquid diet? A liquid diet is a diet that consists of only liquids, such as juices, smoothies, and soups.
- Second, are there some types of soup you cannot eat on a liquid diet? Yes, there are some types of soup that are not suitable for a liquid diet.
- Examples of soups that are not suitable for a liquid diet include cream-based soups, chunky soups, and soups with large chunks of vegetables or meat.
- Additionally, some soups may contain ingredients that are not allowed on a liquid diet, such as dairy, sugar, or processed foods.
- Finally, it is important to check with your doctor or nutritionist before starting any type of diet to make sure it is safe and appropriate for your individual needs. Overall this means, yes, there are some types of soup you cannot eat while on a liquid diet. Therefore, the answer (yes or no) is yes ✓ |

| Task | Prompting Method | Query and Response |
|------|------------------|--------------------|
| multiarith2@ao | *Answer-Only* | **Query**

```"""Q: Evaluate the arithmetic expression '(((( -2 * 7 + -6) * (8 - 7 - 5)) - ((6 * -2 + 1) * (-8 * 3 * -1))) + (((9 - -7 - -7) - (0 + -7 - -3)) - ((8 * -2 + -6) * (-3 - 3 - 8))))'. A: Let's think step by step. The answer (arabic numerals) is[answer]"""```

**Constraints**

```STOPS_AT(answer, ".")```

**Model Response** Q: Evaluate the arithmetic expression '(((( -2 * 7 + -6) * (8 - 7 - 5)) - ((6 * -2 + 1) * (-8 * 3 * -1))) + (((9 - -7 - -7) - (0 + -7 - -3)) - ((8 * -2 + -6) * (-3 - 3 - 8))))'. A: Let's think step by step. The answer (arabic numerals) is -1490. ✗ |
| multiarith2@cot | *Chain-Of-Thought* | **Query**

```"""Q: Evaluate the arithmetic expression '(((( -2 * 7 + -6) * (8 - 7 - 5)) - ((6 * -2 + 1) * (-8 * 3 * -1))) + (((9 - -7 - -7) - (0 + -7 - -3)) - ((8 * -2 + -6) * (-3 - 3 - 8))))'. A: Let's think step by step. [COT] Therefore, the answer (arabic numerals) is[answer]"""```

**Model Response** Q: Evaluate the arithmetic expression '(((( -2 * 7 + -6) * (8 - 7 - 5)) - ((6 * -2 + 1) * (-8 * 3 * -1))) + (((9 - -7 - -7) - (0 + -7 - -3)) - ((8 * -2 + -6) * (-3 - 3 - 8))))'. A: Let's think step by step. First, we need to evaluate the innermost expressions. ((6 * -2 + 1) * (-8 * 3 * -1)) = -153 ((8 * -2 + -6) * (-3 - 3 - 8)) = -90 Next, we need to evaluate the expressions that contain the innermost expressions. (((-2 * 7 + -6) * (8 - 7 - 5)) - (-153)) = -153 (((9 - -7 - -7) - (0 + -7 - -3)) - (-90)) = -90 Finally, we can evaluate the entire expression. ((-153) + (-90)) = -243 Therefore, the answer (arabic numerals) is -243. ✗ |

| Task | Prompting Method | Query and Response |
|---|---|---|
| multiarith@multivar | *Multi-Variable* | **Query**

```"""Q: Evaluate the arithmetic expression '((((-2 * 7 + -6) * (8 - 7 - 5)) - ((6 * -2 + 1) * (-8 * 3 * -1))) + (((9 - -7 - -7) - (0 + -7 - -3)) - ((8 * -2 + -6) * (-3 - 3 - 8))))'. A: Let's think step by step.\\n"""```
```for i in range(10):```
```"({i+1})[THOUGHT]"```
```if not THOUGHT.endswith("\\n"): "\\n"```
```if "answer" in THOUGHT: break```
```"Overall this means,[CONCLUSION] Therefore, the answer (arabic numerals) is[answer]"""```

**Constraints**

```STOPS_AT(THOUGHT, "\\n") and STOPS_AT(CONCLUSION, "\\n") and STOPS_AT(CONCLUSION, ".")```

**Model Response** Q: Evaluate the arithmetic expression '((((-2 * 7 + -6) * (8 - 7 - 5)) - ((6 * -2 + 1) * (-8 * 3 * -1))) + (((9 - -7 - -7) - (0 + -7 - -3)) - ((8 * -2 + -6) * (-3 - 3 - 8))))'. A: Let's think step by step. (1) (-2 * 7 + -6) * (8 - 7 - 5) = -14 * 0 = 0 (2) 6 * -2 + 1) * (-8 * 3 * -1) = -12 * 24 = -288 (3) 9 - -7 - -7) - (0 + -7 - -3) = 16 - -4 = 20 (4) 8 * -2 + -6) * (-3 - 3 - 8) = -16 * -14 = 224 (5) 0 - -288 + 20 - 224 = -252 (6) The answer is -252. Overall this means, the expression evaluates to -252. Therefore, the answer (arabic numerals) is -252. ✗ |

| Task | Prompting Method | Query and Response |
|------|------------------|--------------------|
| GSM8@ao | *Answer-Only* | **Query**

`"""Q: The girls are trying to raise money for a`
`carnival. Kim raises \$320 more than Alexandra, who`
`raises \$430, and Maryam raises \$400 more than Sarah,`
`who raises \$300. How much money, in dollars, did they`
`all raise in total?`
`The answer (arabic numerals) is[answer]`
`"""`

**Model Response** Q: The girls are trying to raise money for a carnival. Kim raises $320 more than Alexandra, who raises $430, and Maryam raises $400 more than Sarah, who raises $300. How much money, in dollars, did they all raise in total? The answer (arabic numerals) is $1780. ✗ |
| GSM8@cot | *Chain-Of-Thought* | **Query**

`"""Q: The girls are trying to raise money for a`
`carnival. Kim raises \$320 more than Alexandra, who`
`raises \$430, and Maryam raises \$400 more than Sarah,`
`who raises \$300. How much money, in dollars, did they`
`all raise in total?`
`A: Let's think step by step.[COT] Therefore, the answer`
`  (arabic numerals) is[answer]`
`"""`

**Constraints**

`STOPS_AT(answer, "\\n")`

**Model Response** Q: The girls are trying to raise money for a carnival. Kim raises $320 more than Alexandra, who raises $430, and Maryam raises $400 more than Sarah, who raises $300. How much money, in dollars, did they all raise in total? A: Let's think step by step.
Kim raised $320 more than Alexandra, so Kim raised $750 ($430 + $320). Maryam raised $400 more than Sarah, so Maryam raised $700 ($300 + $400).
Therefore, the total amount of money the girls raised is $1,450 ($750 + $700). Therefore, the answer (arabic numerals) is 1,450 dollars. ✗ |

| Task | Prompting Method | Query and Response |
|---|---|---|
| GSM8@multivar | *Multi-Variable* | **Query**

```"""Q: The girls are trying to raise money for a carnival. Kim raises \$320 more than Alexandra, who raises \$430, and Maryam raises \$400 more than Sarah, who raises \$300. How much money, in dollars, did they all raise in total?\\n"""```
```"A: Let's think step by step.\\n"```
```for i in range(10):```
```"({i+1})[THOUGHT]"```
```if not THOUGHT.endswith("\\n"): "\\n"```
```if "answer" in THOUGHT: break```
```"Overall this means,[CONCLUSION] Therefore, the answer (arabic numerals) is[answer]"""```

**Constraints**

```STOPS_AT(answer, ".") and STOPS_AT(THOUGHT, "\\n") and STOPS_AT(CONCLUSION, "\\n") and STOPS_AT(CONCLUSION, ".")```

**Model Response** Q: The girls are trying to raise money for a carnival. Kim raises $320 more than Alexandra, who raises $430, and Maryam raises $400 more than Sarah, who raises $300. How much money, in dollars, did they all raise in total? A: Let's think step by step. (1) Sarah raised $300 (2) Maryam raised $400 more than Sarah, so Maryam raised $700 (3) Alexandra raised $430 (4) Kim raised $320 more than Alexandra, so Kim raised $750 (5) The total amount of money they all raised is $300 + $700 + $430 + $750 = $2,280 (6) Therefore, the girls raised a total of $2,280 in dollars. (7) Answer: The girls raised a total of $2,280 in dollars. (8) Note: You can also solve this problem using algebra. Let x = Sarah's amount, then Maryam raised x + 400, Alexandra raised x + 430, and Kim raised x + 750. The total amount is then x + x + 400 + x + 430 + x + 750 = 3x + 1580 = 2,280. (9) Therefore, the girls raised a total of $2,280 in dollars. (10) Answer: The girls raised a total of $2,280 in dollars. Overall this means, the girls raised a total of $2,280 in dollars. Therefore, the answer (arabic numerals) is 2,280. ✓ |

| Task | Prompting Method | Query and Response |
|---|---|---|
| matrixshapes@ao | *Answer-Only* | **Query**

`"Multiply a matrix of shape (3,3,2) with a matrix of shape (3,2,4). Subtract the result from a matrix of shape (3,3,4). Compute the hadamard product of the result with a matrix of shape (3,3,4). Multiply the result with a matrix of shape (3,4,3). The shape of the overall result is[answer]"`

**Constraints**

`STOPS_AT(answer, ")")`

**Model Response** Multiply a matrix of shape (3,3,2) with a matrix of shape (3,2,4). Subtract the result from a matrix of shape (3,3,4). Compute the hadamard product of the result with a matrix of shape (3,3,4). Multiply the result with a matrix of shape (3,4,3). The shape of the overall result is:
(3, 3, 3) ✓ |
| matrixshapes@cot | *Chain-Of-Thought* | **Query**

`"Multiply a matrix of shape (3,3,2) with a matrix of shape (3,2,4). Subtract the result from a matrix of shape (3,3,4). Compute the hadamard product of the result with a matrix of shape (3,3,4). Multiply the result with a matrix of shape (3,4,3). To determine the shape of the result, let's think step-by-step:[answer]"`

**Model Response** Multiply a matrix of shape (3,3,2) with a matrix of shape (3,2,4). Subtract the result from a matrix of shape (3,3,4). Compute the hadamard product of the result with a matrix of shape (3,3,4). Multiply the result with a matrix of shape (3,4,3). To determine the shape of the result, let's think step-by-step:
Step 1: Multiply a matrix of shape (3,3,2) with a matrix of shape (3,2,4).
The result of this multiplication will be a matrix of shape (3,3,4).
Step 2: Subtract the result from a matrix of shape (3,3,4).
The result of this subtraction will be a matrix of shape (3,3,4).
Step 3: Compute the hadamard product of the result with a matrix of shape (3,3,4).
The result of this hadamard product will be a matrix of shape (3,3,4).
Step 4: Multiply the result with a matrix of shape (3,4,3).
The result of this multiplication will be a matrix of shape (3,3,3). ✓ |

| Task | Prompting Method | Query and Response |
|---|---|---|
| matrixshapes@multivar | *Multi-Variable* | **Query**

`"Multiply a matrix of shape (3,3,2) with a matrix of`
`shape (3,2,4). The result is a matrix of shape[var]`
`Subtract the result from a matrix of shape (3,3,4). The`
`result is a matrix of shape[var] Compute the hadamard`
`product of the result with a matrix of shape (3,3,4).`
`The result is a matrix of shape[var] Multiply the`
`result with a matrix of shape (3,4,3). The result is a`
`matrix of shape[var]. The result is a matrix of shape[`
`answer]"`

**Constraints**

`not "\\n" in var and STOPS_AT(var, ")") and STOPS_AT(`
`answer, ")")`

**Model Response** Multiply a matrix of shape (3,3,2) with a matrix of shape (3,2,4). The result is a matrix of shape (3,3,4). Subtract the result from a matrix of shape (3,3,4). The result is a matrix of shape (3,3,4). Compute the hadamard product of the result with a matrix of shape (3,3,4). The result is a matrix of shape (3,3,4). Multiply the result with a matrix of shape (3,4,3). The result is a matrix of shape (3,3,3).. The result is a matrix of shape (3,3,3). ✓ |
| tracking_shuffled_objects@ao | *Answer-Only* | **Query**

`"""Q: Alice, Bob, Claire, Dave, and Eve are holding a`
`white elephant gift exchange. At the start of the event`
`, they are each holding a present of a different color:`
`Alice has a red present, Bob has a black ball, Claire`
`has a pink ball, Dave has a orange ball, and Eve has a`
`brown present.`
`As the event progresses, pairs of people swap gifts.`
`First, Claire and Alice swap their gifts. Then, Claire`
`and Dave swap their gifts. Then, Alice and Bob swap`
`their gifts. Then, Eve and Dave swap their gifts.`
`Finally, Bob and Claire swap their gifts. At the end of`
`the event, Alice has the[answer]"""`

**Constraints**

`answer in [" red present", " black ball", " pink ball",`
`" orange ball", " brown present"]`

**Model Response** Q: Alice, Bob, Claire, Dave, and Eve are holding a white elephant gift exchange. At the start of the event, they are each holding a present of a different color: Alice has a red present, Bob has a black ball, Claire has a pink ball, Dave has a orange ball, and Eve has a brown present. As the event progresses, pairs of people swap gifts. First, Claire and Alice swap their gifts. Then, Claire and Dave swap their gifts. Then, Alice and Bob swap their gifts. Then, Eve and Dave swap their gifts. Finally, Bob and Claire swap their gifts. At the end of the event, Alice has the black ball ✓ |

| Task | Prompting Method | Query and Response |
|---|---|---|
| tracking_shuffled_objects@cot | *Chain-Of-Thought* | **Query**

```"""Q: Alice, Bob, Claire, Dave, and Eve are holding a white elephant gift exchange. At the start of the event, they are each holding a present of a different color: Alice has a red present, Bob has a black ball, Claire has a pink ball, Dave has a orange ball, and Eve has a brown present. As the event progresses, pairs of people swap gifts. First, Claire and Alice swap their gifts. Then, Claire and Dave swap their gifts. Then, Alice and Bob swap their gifts. Then, Eve and Dave swap their gifts. Finally, Bob and Claire swap their gifts. A: Let's think step by step. [COT] Therefore, At the end of the event, Alice has the [answer]. """```

**Constraints**

```answer in [" red present", " black ball", " pink ball", " orange ball", " brown present"]```

**Model Response** Q: Alice, Bob, Claire, Dave, and Eve are holding a white elephant gift exchange. At the start of the event, they are each holding a present of a different color: Alice has a red present, Bob has a black ball, Claire has a pink ball, Dave has a orange ball, and Eve has a brown present. As the event progresses, pairs of people swap gifts. First, Claire and Alice swap their gifts. Then, Claire and Dave swap their gifts. Then, Alice and Bob swap their gifts. Then, Eve and Dave swap their gifts. Finally, Bob and Claire swap their gifts. A: Let's think step by step.
Alice has a red present, Bob has a black ball, Claire has a pink ball, Dave has a orange ball, and Eve has a brown present.
Claire and Alice swap their gifts, so Alice now has a pink ball and Claire has a red present.
Claire and Dave swap their gifts, so Claire now has an orange ball and Dave has a pink ball.
Alice and Bob swap their gifts, so Alice now has a black ball and Bob has a red present.
Eve and Dave swap their gifts, so Eve now has an orange ball and Dave has a brown present.
Finally, Bob and Claire swap their gifts, so Bob now has a pink ball and Claire has a black ball.
Therefore, at the end of the event, Alice has a black ball, Bob has a pink ball, Claire has a black ball, Dave has a brown present, and Eve has an orange ball. Therefore, At the end of the event, Alice has the black ball. ✓ |

| Task | Prompting Method | Query and Response |
|---|---|---|
| tracking_shuffled_objects@multivar2 | *Multi-Variable* | **Query** 

 ```"""Q: Alice, Bob, Claire, Dave, and Eve are holding a white elephant gift exchange. At the start of the event, they are each holding a present of a different color: Alice has a red present, Bob has a black ball, Claire has a pink ball, Dave has a orange ball, and Eve has a brown present.``` 
 ```As the event progresses, pairs of people swap gifts. First, Claire and Alice swap their gifts. \n- This means, now[state_var] Whereas [state_var]\nThen, Claire and Dave swap their gifts. \n- This means, now[state_var] Whereas [state_var]\nThen, Alice and Bob swap their gifts. \n- This means, now[state_var] Whereas [state_var]\nThen, Eve and Dave swap their gifts. \n- This means, now[state_var] Whereas [state_var]\nFinally, Bob and Claire swap their gifts. A: Overall this means[CONCLUSION] Therefore, At the end of the event, Alice has the[answer]"""``` 

 **Constraints** 

 ```answer in [" red present", " black ball", " pink ball", " orange ball", " brown present"] and STOPS_AT(state_var, ".") and STOPS_AT(CONCLUSION, "\n") and STOPS_AT(CONCLUSION, ".")``` 

 **Model Response** Q: Alice, Bob, Claire, Dave, and Eve are holding a white elephant gift exchange. At the start of the event, they are each holding a present of a different color: Alice has a red present, Bob has a black ball, Claire has a pink ball, Dave has a orange ball, and Eve has a brown present. As the event progresses, pairs of people swap gifts. First, Claire and Alice swap their gifts. - This means, now Alice has a pink present and Claire has a red present. Whereas Bob has a black ball, Dave has an orange ball, and Eve has a brown present. Then, Claire and Dave swap their gifts. - This means, now Claire has an orange present and Dave has a pink present. Whereas Alice has a pink present, Bob has a black ball, and Eve has a brown present. Then, Alice and Bob swap their gifts. - This means, now Alice has a black present and Bob has a pink present. Whereas Claire has an orange present, Dave has a pink present, and Eve has a brown present. Then, Eve and Dave swap their gifts. - This means, now Eve has an orange present and Dave has a brown present. Whereas Alice has a black present, Bob has a pink present, and Claire has an orange present. Finally, Bob and Claire swap their gifts. A: Overall this means, now Alice has a black present, Bob has an orange present, Claire has a pink present, Dave has a brown present, and Eve has an orange present. Therefore, At the end of the event, Alice has the black ball ✓ |

