# OpenReview forum: "Prompt Sketching for Large Language Models"
_ICLR.cc/2024/Conference — Submitted to ICLR 2024_

### Official Review · Reviewer_pDQp · 2023-10-30

**Soundness:** 2 fair
**Presentation:** 2 fair
**Contribution:** 2 fair
**Rating:** 5
**Confidence:** 4

**Summary:**

This paper proposes prompt sketching, a method to first provide sketches to the language model, then ask the model to fill in certain variables. The authors did experiments on several reasoning tasks and some planning tasks (with state tracking), to show the proposed method outperform existing method like direct prompting and chain-of-thought prompting. The models used are InstructGPT-based (text-davinci-003) and Llama-2 Chat based.

**Strengths:**

- The motivation of this paper is great, and the sketching idea is highly interesting. Currently most language models do decoding in an auto-regressive fashion and might not adhere to certain constraints in the input. Sketching can definitely help models better plan and output responses better fit into user constraints.

- Some of the tasks explored are quite novel and interesting, like the interleaved reasoning tasks and the planning tasks (section 4.2), and the experiments do show they benefit from prompt sketching quite a bit.

**Weaknesses:**

The biggest concern is the experiments in this paper, which do not clearly show the benefits of the proposed method:

Most of the explored tasks, including logical reasoning, question answering, and arithmetic reasoning, use the *multi-variable* prompting method (BeamVar, Var) as the sketch (Figure 3), which is actually a variant of the self-consistency [1] method: sample multiple chain-of-thoughts and then aggregate. Hence a fair comparison should be between the proposed method and self-consistency-based chain-of-thought, under the exact same number of samples.
- The novelty of the proposed method compare to self-consistency should be discussed in details in this paper.
- Can the authors add self-consistency with the same number of samples as a baseline?
- Comparing chain-of-thought prompting under BeamVar and prompt-sketching under BeamVar (this should be a more fair comparison with the same number of sampled thoughts), the proposed method does not yield much gains. Hence the authors should better discuss what is the main contribution of "sketching" over existing chain-of-thought.

[1] Wang et al. Self-Consistency Improves Chain of Thought Reasoning in Language Models. ICLR 2023.

In section 4.2, some novel tasks are explored and could potentially show the benefits of the proposed sketching. However, the experiments are extremely small-scale (10 Sudoku puzzles, 10 Dungeon environments), so it is unclear whether the proposed method indeed outperform existing methods.

Performance gains: from Table 6, the confidence intervals are fairly large, and it is unclear which method is significantly better compared to the others. Can the authors clarify which result is statistically significant?

Computational cost: can the authors discuss in more details on the exact computational cost used for the proposed method?

**Questions:**

- Can the authors add self-consistency with the same number of samples as a baseline?
- Table 6, the confidence intervals are fairly large, and it is unclear which method is significantly better compared to the others. Can the authors clarify which result is statistically significant?
- Computational cost: can the authors discuss in more details on the exact computational cost used for the proposed method?

---

> ### Author Response · Authors · 2023-11-21
> **Response to Reviewer pDQp Part I**
>
> We thank the reviewer for their insightful questions, which we address below, and are encouraged to hear that they find prompt sketches to be a highly interesting idea.
>
> **Q: How does Sketching with BeamVar/Var relate to Self-Consistency and can you add it as a baseline?**
> The self-consistency [1] decoding strategy improves LLM reasoning by greedily sampling multiple (chain-of-thought) reasoning paths and aggregating the resulting final answers, by (unweighted or weighted) majority vote. In contrast, sketch-aware decoders generate answers token-by-token or thought-by-thought (not path by path), can branch on the top-n alternatives at each step, and do not aggregate the final results. Instead, a single sequence is chosen as the one answer to an input, considering the model likelihood of the underlying token sequence only (no aggregation or majority voting). With this, self-consistency and sketching can be considered orthogonal approaches and could even be applied jointly, i.e., sample-decode a sketch multiple times and aggregate the results.
> We added a comparison of sketching and self-consistency to the revised manuscript (Appendix D). For this comparison, we choose $n=2$/$n=4$ samples for self-consistency, as this aligns with the computational overhead of our branching decoders BeamVar/Var with $n=2$. However, our results show that, in many cases, self-consistency with such a low number of samples does not even outperform argmax CoT which is the main baseline we outcompete with sketching. This aligns with the self-consistency paper itself [1], which reports that around 5-20 consistency samples are required to outperform simple argmax CoT. We thus find that within this compute budget ($n=2$/$n=4$), argmax CoT and thus sketching both outperform self-consistency decoding.
>
> **Q: Is CoT+BeamVar a better baseline for Sketching+BeamVar?**
> Comparing CoT+BeamVar with Sketching+BeamVar, we observe an accuracy increase of $8-12%$ with OpenAI models for AQuA, StrategyQA, Tracking Shuffled Objects and Matrix Shapes. On these tasks, CoT+BeamVar is thus clearly outperformed by sketching. For Date Understanding, Multistep Arithmetic, GSM8K on the other hand we indeed observe comparable performance with sketching+BeamVar. This can be explained by the two-part nature of the zero-shot CoT formulation [2] (first decoding reasoning and then the final answer). Applying a sketch-aware decoder like BeamVar to this kind of prompt already boosts performance, as BeamVar can exploit this two-variable structure. For a baseline comparison (no interaction between multiple variables), we thus have to look at Beam+CoT (one column to the left). There, we find that Multistep Arithmetic and GSM8K again perform worse. Only for Date Understanding do sketching and sketch-aware decoders not appear to bring particular benefit for OpenAI models, which is reflected in our overall summary: sketching outperforms sequential prompting on 7/8 tasks.
>
> **Q: Can you extend your evaluation of additional applications and further demonstrate the effectiveness of sketching in this context?**
> We have scaled our experiments on the additional applications (Section 4.2) by a factor of 10 and now evaluate a larger number of samples (100 samples per model/decoder configuration). We report the results in the updated manuscript. Overall, we observe comparable trends to before, i.e., sketch-aware decoders clearly show the ability to backtrack in Sudoku solving and solve the Dungeon Escape tasks more efficiently (fewer steps), when compared to naive argmax decoding. We have also added an additional experiment on TextWorld [3] exploration, which leverages our decoders for LLM-guided world exploration. Again, we find that sketch-aware decoders clearly outperform greedy argmax decoding, completing TextWorld quests of different lengths with 20% fewer steps. We further discuss our TextWorld results in Appendix B.2.
>
> **References**
> [1] Self-consistency improves chain of thought reasoning in language models, X. Wang et al., ICLR’23
> [2] Large language models are zero-shot reasoners, K. Takeshi et al., NeurIPS’22
> [3] Textworld: A learning environment for text-based games, M.-A. Côté et al., IJCAI’18

---

> > ### Author Response · Authors · 2023-11-21
> > **Response to Reviewer pDQp Part II**
> >
> > **Q: Can you clarify which results are statistically significant given the evaluation sample sizes and  fairly large confidence intervals in Table 6?**
> > We understand your concerns regarding sample size and statistical significance for the OpenAI-specific part of our evaluation. Unfortunately, we cannot scale our OpenAI experiments further, due to cost constraints ($4000 for the results at the current scale). However, we have expanded our Llama-2-based experiments (Table 2) to include all decoder/dataset configurations and evaluate 10 times as many samples as in the OpenAI experiments (1000 per dataset, or the full datasets in many cases) and also provide confidence intervals for the Llama results in the new appendix Table 8. In this much more scaled Llama-2 setting, we observe comparable trends to the smaller-scale OpenAI experiments: Across tasks, sketching improves overall reasoning performance when compared to sequential prompting, while sketch-aware decoders can provide further gains in exchange for higher computational cost.
> >
> > **Q: Can you discuss the computational cost of the proposed method in further detail?**
> > Computationally, argmax sketching is cheaper than sequential decoding for the same output. This is because in sketching the model does not have to autoregressively process fixed template parts (assuming sufficient model access and key-value caching). For BeamVar the computational cost is increased by a factor of beam width $n$, whereas Var requires a factor of $n^2$ more sequences to be explored (2x and 4x, respectively, for our main evaluation). This is a considerable computational overhead, but, as known from traditional beam search, can be worth the additional compute if downstream performance is absolutely crucial. It is also more cost-effective than, e.g., self-consistency decoding, which requires 5-20 consistency samples, resulting in a 5-20x overhead. In practice, this kind of performance/accuracy trade-off of course depends on the concrete use case, compute budget, and user requirements.
> >
> > We hope to have been able to address all the reviewers’ concerns, are happy to answer any follow-up questions they might have, and are looking forward to their reply.

---

> > > ### Comment · Reviewer_pDQp · 2023-11-22
> > >
> > > Thanks for the clarification. I agree that for Interleaved Reasoning, the win is pretty clear and the underlying reason is clear as well (the model does more explicit state tracking via the sketches), however, for other tasks, the wins are fairly small (and seem to vary a lot based on which one of argmax/beamvar/var/beam is used).
> > >
> > > For math tasks like AQuA or GSM8K, can the authors clarify exactly why sketching would help? From the prompts it seems like it's only forcing two things in addition to CoT:
> > > (1) the number of thought steps: how is this determined btw? I saw 12 is used for AQuA and 10 is used for GSM8K, are they randomly chosen? It's possible that by enforcing a certain number of thought steps, the model generates a *longer* thought process which leads to some of the gains. Can the authors provide some simple statistics like the avg # tokens in CoT and in sketching?
> > > (2) some formatting on the thoughts: this issue can actually easily be addressed by few-shot CoT (i.e., by showing models a few examples, the model can better adhere to the few-shot format). From Table 4 we can also see the gains become much smaller when few-shot is used.

---

> > > > ### Author Response · Authors · 2023-11-22
> > > >
> > > > Thanks for engaging in the discussion! We are happy to hear that the reviewer agrees with us on the clear advantages of sketching for tasks that can be framed as interleaved reasoning.
> > > > Regarding the math tasks, we agree with the reviewer that both enforcing the reasoning format and increasing the reasoning chain length are likely to contribute to the performance improvement we observe. In addition, we believe that sketch-aware decoders effectively back-track and correct reasoning errors (see the improvement from 0.32 for argmax to 0.35 with Var and BeamVar, all with sketching, on GSM8K in Table 4).
> > > >
> > > > In more detail:
> > > > While our templates permit 12 and 10 reasoning steps, these are upper bounds that the model can break from earlier (by producing a corresponding stopping phrase). We based these limits on experimentation with example data, which showed that naturally, the model would very rarely exceed these bounds in well-formed reasoning processes. In practice, for sketched argmax, we observe on average 6.77 and 8.29 thought steps for AQuA and GSM8K, respectively. We also inspected the resulting model output at the step limit, and could not find any instances of truncated reasoning, where the model did not conclude with its final answer. Compared to non-sketched chain-of-thought, we observe ~20% longer reasoning chains, which can indeed affect performance.
> > > > Regarding the results presented in Table 4 (revised draft): Indeed, in the few-shot setting, performance improvements are reduced as the advantageous formatting enforced by sketching can also be encouraged using few-shot prompting. However, our sketch-aware decoders combined with sketching applied in a 0-shot setting already outperform argmax decoding without sketching in the two-shot setting on all four considered tasks. We conclude that the rigorous formatting guidelines enforced by sketching can replace and even improve upon the soft ones imposed by few-shot prompting, crucially, without requiring the few-shot demonstrations which come with important drawbacks:
> > > > * *Prompt length:* Including few-shot examples increases the number of input tokens and thereby latency and cost.
> > > > * *Handcrafting examples:* Few-shot prompting needs examples in order to be applied to a new task. which need to be manually crafted. In contrast, sketching only requires the often transferable prompt sketch, which can be adapted (if needed) at the same effort as adapting a prompt.
> > > >
> > > > We hope this addresses your follow-up questions. Should you want any further clarification, please don’t hesitate to ask.

---

> > > > > ### Comment · Reviewer_pDQp · 2023-11-22
> > > > >
> > > > > Thanks for the further clarification. Overall I think the idea proposed in this paper is novel and interesting, and the gain on interleaved reasoning is clear, hence I will increase my score.
> > > > >
> > > > > However, based on the responses I think this paper should better clarify its contribution on all the other reasoning tasks:
> > > > > 1) the proposed example reasoning framework ("on one hand.., on the other hand") is not generally adopted for all tasks (only used in "Information Essentiality") and it is unclear why it could be beneficial on top of CoT. It also seems to be task-specific (e.g., some tasks require thinking both pros and cons) and needs to be manually designed by looking at task examples;
> > > > > 2) most sketches adopt the "for i in range(K): -THOUGHT" framework, but it seems like multiple factors could contribute to the final gains: prompt length, formatting, backtracking, rather than just "sketching" which is the main claim of this paper. The number K also needs to be manually selected by looking at examples and model responses.
> > > > > I think this paper could be further improved by better clarifying its contributions, analysis on what actually contributes to the gains compared to existing methods, and more clear presentation on the methods/prompt used.

---

> > > > > > ### Author Response · Authors · 2023-11-22
> > > > > >
> > > > > > Thank you for your response and analysis.
> > > > > >
> > > > > > We agree with the reviewer that investigating the exact mechanisms leading to optimal performance on different reasoning tasks is an exciting and likely fruitful research direction for follow-up work. However, in this work, we want to establish the foundation for such investigations by showing, for the first time, that structural guidance of the LLM reasoning process can improve its capabilities and provide the conceptual and tooling framework to rigorously investigate related questions. We believe this to be of high practical importance as first tools enabling such structural guidance [1,2] are rapidly gaining popularity (jointly over 17k GitHub stars).
> > > > > >
> > > > > > In particular, sketching in combination with sketch-aware decoders is exactly what enables constraints on reasoning chain length, formatting, and backtracking. However, we intentionally focus on simple sketches (with potentially suboptimal hyper-parameters) and a general analysis instead of exploring more complex or even grammar-guided prompt sketches, as our goal is to establish whether structural support can improve model reasoning capabilities at all. Thus, we believe that, while our sketching framework and decoders support these more complicated forms of template-guided inference, such investigations are out of scope for this work and not permitted by the rigorous space constraints of a single ICLR paper.
> > > > > >
> > > > > > We believe that the mathematical reasoning problems discussed above are an interesting case study and are happy to include them in the appendix.

---

### Official Review · Reviewer_tG6m · 2023-10-30

**Soundness:** 3 good
**Presentation:** 3 good
**Contribution:** 2 fair
**Rating:** 6
**Confidence:** 3

**Summary:**

This paper proposes a novel prompting method (Prompt Sketching) which guides intermediate inference steps based on template. Prompt Sketching provides more control over generation and inference steps by putting deterministic chunks in decoding steps. In addition to the prompting strategy, authors suggests two variants of beam search (VAR, BEAMVAR) to adapt LLM to new decoding process. Experiments show its effectiveness in LLM reasoning tasks over CoT. Also, authors suggests types of task for which prompt sketching can be especially useful.

**Strengths:**

- Simple prompting strategy to improve LLM reasoning performance
- New applications are interesting and could be useful for launching practical AI services.
- Structured outputs induced by prompt sketching have potential to automate various tasks beyond suggested applications.
- The suggested method can reduce the number of model calls compared to stop-and-go and thus reduce the cost, which is practical.

**Weaknesses:**

- Generating templates requires human intervention and may necessitate significant efforts until finding a template working well. Also, potentially, templates can overfit to evaluation datasets.
- It does not work well for small LMs.
- Evaluation results are given with limited amounts of data, which may harm the credibility of the results. Especially, confidence intervals in Table 6 look pretty large.
- Most of new applications look already doable by guidance-ai (https://github.com/guidance-ai/guidance ), which is cited in the paper. Also, naive stop-and-go is not compared in main results.

**Questions:**

- What’s the Sequential Prompting used in Table 3? CoTs or stop-and-go?
- Can templates be generated or suggested by LLM as well? I am also wondering if templates can be generated by retrieval.
- Is the suggested method applicable to programming tasks?
- Can Prompt sketching get help from demonstrations?

---

> ### Author Response · Authors · 2023-11-21
> **Response to Reviewer tG6m Part I**
>
> We thank the reviewer for their insightful questions, which we address below, and are encouraged to hear that they find prompt sketches to be an intuitive format for chained queries.
>
> **Q: Can you comment on the overhead of writing sketches as opposed to prompts?**
> Similar to prompt engineering, templates have to be constructed, tested, and debugged with example data to ensure the best performance. With current models, however, this prompt/template engineering effort cannot be avoided, as a single input/sentence is typically not sufficient to ensure that the model conforms to very specific and precise requirements. In this context, we consider sketching a more controlled form of prompting, where the output format is guaranteed and not up to prompt engineering and randomness in the generation process. Nonetheless, we agree that a human effort has to be made to properly prompt, template, and use LLMs. See also Section 4.3 for more discussion of this aspect.
>
> **Q: Is prompt-sketching less effective with smaller LLMs?**
> We note that even with a smaller Llama-2 Chat model (13B parameters), sketching outperforms sequential prompting on 6/8 reasoning tasks while exhibiting even greater performance gains than for large OpenAI models (up to 8% vs. up to 4%). However, as we acknowledge in our evaluation, Llama-2 seems incapable of solving the Matrix Shape and AQuA reasoning tasks, irrespective of the prompting/sketching scheme, leading to inconclusive results on these datasets. Other tasks are not as strongly affected and show comparable trends to the OpenAI results. Further, we demonstrate in Appendix B.1 that sketching can be very effective with smaller models, for instance when asking a model to produce valid JSON output: For the small text-curie-001, sketching allows us to guarantee valid JSON output, whereas the same model with a corresponding prompts, is unable to produce any valid JSON at all.
>
> Potentially, our description in Section 4 was unclear in this regard. We have updated it in the rebuttal revision to make the above points more clear.
>
> **Q: Does the sample size in the evaluation impact the credibility of the results?**
> We understand your concerns regarding sample size and statistical significance for the OpenAI-specific part of our evaluation. Unfortunately, we cannot scale our OpenAI experiments further, due to cost constraints ($4000 for the results at the current scale). However, we have expanded our Llama-2-based experiments (Table 2) to include all decoder/dataset configurations and for 10 times as many samples as in the OpenAI experiments (1000 per dataset, or the full datasets in many cases). We also provide confidence intervals for the Llama results in the new appendix Table 8. In this much larger-scale Llama-2 setting, we observe comparable trends to the smaller-scale OpenAI experiment: Across tasks, sketching improves overall reasoning performance when compared to sequential prompting, while sketch-aware decoders can provide further gains in exchange for higher computational cost.
>
> **Q: How does the presented approach differ from Guidance?**
> Indeed, practical tools like Guidance also enable a form of stop-and-go inference, which can be seen as a simple form of sketching. However, Guidance is an ad-hoc approach, where templates are simply decoded using multiple isolated LLM calls. In contrast, sketching theoretically anchors this approach as a multi-part sequence decoding problem, enabling sketch-aware decoders. Further, prior work (including Guidance) does not provide insights on whether this form of structural guidance positively impacts model reasoning capabilities. This is evaluated as the argmax version of prompt sketching. Our experimental results are thus also of interest to Guidance-like frameworks, as we show that structural guidance during inference can indeed improve reasoning capabilities, beyond syntax and structure. To the best of our knowledge, this has not been shown before and therefore improves the understanding of simple stop-and-go inference in general. We have updated the introduction of the paper to better reflect this.

---

> > ### Author Response · Authors · 2023-11-21
> > **Response to Reviewer tG6m Part II**
> >
> > **Q: What’s the sequential prompting used in Table 3?**
> > For Table 3, we reorder the prompt to include the Sudoku puzzle with missing values and then ask the model to repeat the template with values filled in. This works for simple templates, however does not support dynamically constructed templates or interactive environments.
> >
> > **Q: Could templates be generated or suggested by LLMs?**
> > We have implemented a prototype system that uses an LLM to produce templated sketches that subsequently can be executed using our decoders. However, we have not yet completed comprehensive empirical experiments on the effectiveness of this system. Early results show that a few-shot prompted LLM can easily adopt the sketching format, and indeed produce valid sketches with custom stopping phrases and constraints.
> >
> > **Q: Is sketching applicability to programming tasks?**
> > Sketching is agnostic with respect to the concrete templated data. For instance, sketching and sketch-aware decoders could be used for in-filling problems in program synthesis. Program sketching [1] is a common paradigm in program synthesis and could be implemented using prompt sketching.
> >
> > **Q: Does Sketching benefit from (few-shot) demonstrations?**
> > Sketching and few-shot prompting are orthogonal strategies, i.e., they can be applied independently and jointly. We provide results on combining both in Appendix C.1. Overall, we find that few-shot prompting improves performance across the board, however, sketching still outperforms sequential prompting and, in some cases, zero-shot sketching even outperforms two-shot sequential prompting, suggesting that sketching could even serve as a replacement for few-shot demonstrations.
> >
> > We hope to have been able to address all the reviewer’s concerns, are happy to answer any follow-up questions they might have, and are looking forward to their reply.
> >
> > **References**
> > [1] "Program sketching." A. Solar-Lezama, International Journal on Software Tools for Technology Transfer 15 (2013)

---

> > > ### Comment · Reviewer_tG6m · 2023-11-22
> > >
> > > Thank you for the detailed response! Since clarifications nicely addressed my concerns, I have raised my score to 6.

---

### Official Review · Reviewer_Apoy · 2023-11-01

**Soundness:** 3 good
**Presentation:** 3 good
**Contribution:** 2 fair
**Rating:** 6
**Confidence:** 3

**Summary:**

This paper proposes templated prompt sketches for problems requiring structured generation from LLMs. Structurally constrained generation is an important but overlooked problem. The paper also proposes sketch-aware decoding that considers the structured variables in decoding, and releases the code as an open-source library.

**Strengths:**

The motivation is clear, and the proposed methods which performs beamsearch over the variables (to be generated) is reasonable.

A thorough study of non-templated and stop-and-go method as well as the proposed method, using various decoding strategies, is provided.

The provided prompt sketches are useful for various tasks.

**Weaknesses:**

The experiments show that stop-and-go inference works well, and the proposed method does not significantly improve performance despite the additional overhead. Further, on many of the tasks simple autoregressive CoT seems sufficiently close in performance.

While the paper provides some additional applications for prompt sketches, the tasks and the performance on the tasks are not entirely convincing.

**Questions:**

1. How is the custom decoding applied when using OpenAI API?
2. I'm curious about the results if few-shot prompts are used.

---

> ### Author Response · Authors · 2023-11-21
> **Response to Reviewer Apoy**
>
> We thank the reviewer for their encouraging response (clear motivation, thorough empirical study, useful for various tasks) and insightful observations and reply to their queries below.
>
> **Q: Can you comment on the effectiveness of sketching/sketch-aware decoders with different tasks and models, and how it compares to argmax+CoT?**
> We consider the contribution of our experimental results to be twofold:
> * We demonstrate that template-guided inference via simple argmax-decoded sketches (stop-and-go) and comparatively simple structural templates can already improve task accuracy on reasoning tasks (OpenAI: 7/8, Llama-2: 6/8) significantly (4% and 8% respectively). To the best of our knowledge, this has not been demonstrated before, as existing work on template-guided inference does not discuss the impact on reasoning performance. We thus, for the first time, empirically validate the effectiveness of template-guided (stop-and-go) inference on reasoning tasks, using our sketching framework. Computationally, argmax sketching is cheaper than sequential decoding for the same output (e.g., argmax+CoT), as in sketching/stop-and-go, the model does not have to autoregressively process fixed template parts (assuming sufficient model access and key-value caching). In comparison, simple sketching thus provides accuracy improvements without necessarily incurring any computational overhead (potentially even reducing it).
> * We also demonstrate that while argmax sketching/stop-and-go inference can be effective, sketch-aware decoding brings even further improvements. These decoders incur computational overhead, but, as with traditional beam search, can be worth the additional compute if downstream performance is absolutely crucial. Of course, in practice this trade-off depends entirely on the concrete use case, compute budget, and user requirements.
>
> **Q: Can you extend your evaluation of additional applications and further demonstrate the effectiveness of sketching in this context?**
> We have scaled our experiments regarding additional applications by a factor of 10, and now evaluate a larger number of samples (100 samples per model/decoder configuration). We report the results in the updated manuscript in Table 3. Overall, we observe comparable trends to before, i.e., sketch-aware decoders clearly show the ability to backtrack in Sudoku solving and solve the Dungeon Escape tasks more efficiently (fewer steps). We have also added an additional experiment on TextWorld [1] exploration (Table 3), which leverages our decoders for LLM-guided world exploration. Again, we find that sketch-aware decoders clearly outperform greedy argmax decoding, completing TextWorld quests of different lengths with ~20% fewer steps. We discuss our TextWorld result in appendix B.2.
>
> **Q: How is custom decoding possible with the OpenAI API?**
> The proposed decoders only require rather minimal support by the API/model. In particular, we require the possibility to extend all of the sequences in the beam and obtain logprobs for these sequences for at least their top $n$ continuations. For the OpenAI Completion API models we use in the paper, these capabilities are available.
>
> **Q: How does sketching perform when combined with few-shot prompting?**
> Sketching and few-shot prompting are orthogonal strategies, i.e., can be applied independently and jointly. We provide results on combining both in Appendix C.1. Overall, we find that few-shot prompting improves performance across the board, however, sketching still outcompetes sequential prompting, and in some cases zero-shot sketching even outperforms two-shot sequential prompting, suggesting that sketching can even serve as a replacement for few-shot demonstrations.
> We hope to have been able to address all the reviewers’ concerns, are happy to answer any follow-up questions they might have, and are looking forward to their reply.
>
> **References**
> [1]  Textworld: A learning environment for text-based games, M.-A. Côté et al., IJCAI’18

---

### Official Review · Reviewer_AyRy · 2023-11-01

**Soundness:** 3 good
**Presentation:** 3 good
**Contribution:** 2 fair
**Rating:** 5
**Confidence:** 4

**Summary:**

This paper proposes a new approach to decoding LLM outputs when chaining multiple LLM queries. Such chains of queries can be specified as *sketches*: natural-language prompts that contain *holes* that the LLM is meant to fill in. Each hole is associated with a stopping phrase, and a natural way to read the sketch is as specifying an interaction pattern, where we alternate between (1) deterministically extending an LLM's context window with the next (non-hole) chunk of the sketch, and (2) allowing the LLM to fill in the value of a hole by sampling tokens freely until it emits the stopping phrase for that hole. Because LLMs are autoregressive, this interaction pattern does not allow the LLM propose values for the holes in a way that is *aware* of future interactions in the sketch. To alleviate this problem somewhat, the paper presents two new decoding algorithms (variants of beam search) that optimize the joint log probability of the entire LLM context. On several benchmark tasks, the paper compares the zero-shot performance of LLMs with standard prompts + standard decoding algorithms, vs. with particular prompt sketches and the new proposed decoding algorithms.

**Strengths:**

* Prompt sketches are an intuitive format for specifying certain types of chained queries.

* The paper identifies a connection between decoding for these prompt sketches and constrained decoding, and points out (correctly) that standard beam search is insufficient for this task. The variants of beam search that the paper introduces are largely sensible, and overcome the key barriers to performing beam search in the multi-variable setting—namely, the fact that beams with the same number of tokens may be at different stages of the sketch, making their scores difficult to compare fairly.

* Results are reported both for open models and closed (OpenAI) models. Many souped-up decoding methods require information that is not available through the OpenAI APIs, and it's nice that the authors have shown that a version of their approach (at least for small beam sizes) *can* be implemented atop the user-facing API (at least for text-davinci-003).

**Weaknesses:**

* I couldn't quite follow the motivation: what problem with existing decoding techniques is prompt sketching meant to address? Figure 1 comes closest to illustrating the problem, but it was not particularly compelling. (I am not sure which model was used to generate Figure 1, but I tried copying the prompt and constraint into text-davinci-003 and it had no trouble following the constraint.) To be sure, there are many sketches that I am sure GPT-3 would often fail to follow, even if the sketch were included in the prompt; you can encode arbitrarily difficult infilling problems into sketches. But the sketches presented in this paper are enforcing very simple formatting constraints on, e.g., the list of thoughts generated for a chain-of-thought prompt. What failure modes do you see when just explaining to the model that it should follow the desired format (e.g. by pasting the sketch into the context)? Do failures to follow the format cause failures in reasoning? How exactly do VAR and BEAM_VAR address these failures? (Can they really be doing much, at a beam size of only n=2?)


* The experiments provide somewhat weak evidence for the value of the new decoding methods. In different tasks, it often seems to be the case that *one* of the methods outperforms an argmax baseline, whereas the *other* method underperforms the baseline, and which method wins varies from task to task. Even when the new decoding methods provide a modest advantage over argmax decoding, it is not clear whether the advantage is worth the added computational cost (or dollar cost, for OpenAI models).


* I am not convinced the experimental setup is completely debugged. For example, in chain-of-thought for the "date understanding" task, a period is used as the stopping phrase for each thought. However, periods show up frequently in dates (e.g., "Sept. 1"), and this stopping-phrase is clearly causing the model to cut off thoughts early (page 21). Some experimental settings are also missing details; e.g., in the single-variable chain-of-thought prompts, it is unclear when the [COT] variable ends -- I did not see a stopping phrase reported.


* Some of the algorithmic choices in VAR / BEAM_VAR were not sufficiently justified, and struck me as slightly odd. For example, the VAR algorithm shrinks the n^2 generations for a variable back down to a beam width of n *before* adding the next deterministic chunk. But I thought a key point of these algorithms was to enable the next deterministic chunk to provide a "score" for the proposed variable values; wouldn't it make more sense to rank all n^2 variable proposals by how well they fit with the next deterministic chunk, scale back down to n, and then generate proposals for the next variable?

**Questions:**

I'd appreciate your thoughts on the questions raised in the "weaknesses" section. In particular, it would be great to better understand example failure modes of simpler methods (e.g., argmax decoding for few-shot chain-of-thought prompting) and how prompt sketching addresses / avoids these failures.

---

> ### Author Response · Authors · 2023-11-21
> **Response to Reviewer AyRy Part I**
>
> We thank the reviewer for their insightful questions, which we address below, and are encouraged to hear that they find prompt sketches to be an intuitive format for chained queries.
>
> **Q: Can you clarify the motivation for sketching and provide failure modes of existing models/decoding methods?**
> As pointed out by the reviewer, template-guided decoding can also be implemented by simply showing the LLM a template and asking it to produce a corresponding response. However, there are several limitations and failure modes with this approach:
> - *Instruction Following Capabilities:* Only big and powerful models are currently capable of this kind of instruction-following (e.g., see how `text-curie-001` fails for this kind of task in Table 3, Sudoku). In contrast, sketching also works with small models, which may not understand the template, but can still produce useful completions for individual template variables.
> - *Prompt Size:* Including large templates fully in the prompt will greatly increase the number of required tokens, possibly even exceeding the context window of the model. Further, LLM retrieval across large inputs remains difficult, which can lead to deviations from the provided template. In contrast, sketching guides the model through the template, step by step, without increasing the total number of tokens or requiring long-range retrieval.
> - *Accuracy, Reliability, and Automation:* Instruction-following models are still not perfect, which means templates may be violated and not followed strictly. In contrast, sketching provides a 100% guarantee that the template will be adhered to. This is crucial in automated settings, where the LLM’s output format must always be correct to enable processing by other systems that expect a specific format (e.g., function calling).
> - *Dynamic Templates and Interactive Environment:* In some cases, providing the full template may also not be possible, as it may not be known fully ahead of time. For instance, templates may be dynamic, e.g., multiple repetitions of some template part, until a certain variable value is produced. Models can be instructed to respect such constraints, but this kind of prompting ultimately remains unreliable while sketching can enforce these properties strictly (cf. sketched CoT in the paper). Further, sketching is also applicable to interactive settings, where the template depends on and reacts to model output. For instance, in Table 3 we experimented with Dungeon-style graph traversal (Table 3), and in the updated revision, we added an additional experiment on TextWorld exploration.
>
> **Q: How exactly do BeamVar/Var address these failure modes (with $n=2$)?**
> As pointed out above, sketching alone already greatly alleviates these failure modes as it leads to strict adherence to the template and allows for scripted follow-up. Sketch-aware decoders (BeamVar/Var) additionally allow us to jointly score multiple variables (while standard decoders only allow the last variable to be scored conditioned on all preceding ones). This enables picking variables based on later sequences in the template, which is particularly important in dynamic environments where the template text depends on the value of prior variables. While modern RLHF models are very strong and typically assign high probabilities to the top-1 token, a small beam width of n=2, can already be useful to track an alternative hypothesis when ambiguities arise (demonstrated by our experiments on reasoning tasks). For our more advanced experiments with other applications (cf. Table 3), we apply BeamVar and Var with higher beam width (n=5 and n=3 respectively), as clarified in the revised draft, which enables the exploration of a broader hypotheses space.
>
> **Q: Why are you focusing on relatively simple formatting constraints?**
> We chose to focus on simple sketches, to examine whether structural support can improve model reasoning capabilities at all. Indeed, more complex or even grammar-guided approaches are possible, however, the design space of such sketches/reasoning grammars is very large and requires significant prompt engineering. In contrast, the main goal of this research is to establish foundational insight on whether structural guidance of the LLM reasoning process can improve its capabilities at all. Nonetheless, our sketching framework and decoders are general and capable of more complicated forms of template-guided inference, setting up future work to explore this direction.

---

> > ### Author Response · Authors · 2023-11-21
> > **Response to Reviewer AyRy Part II**
> >
> > **Q: Can you comment on the effectiveness of your methods with different tasks and models? Is it worth the computational overhead?**
> > We consider the contribution of our experimental results to be twofold:
> > * We demonstrate that template-guided inference via simple argmax-decoded sketches and comparatively simple structural templates can already improve task accuracy on reasoning tasks (OpenAI: $7/8$, Llama-2: $6/8$) greatly ($4%$ and $8%$ respectively). To the best of our knowledge, this has not been demonstrated before, as existing work on template-guided inference does not focus on the impact on reasoning performance. Computationally, argmax sketching is also cheaper than sequential decoding for the same output, as in sketching, the model does not have to autoregressively process fixed template parts (assuming sufficient model access and key-value caching).
> > * We also demonstrate that while argmax sketching/stop-and-go inference can be effective, sketch-aware decoding brings even further improvements. For Beam-Var the computational cost is increased by a factor of beam width $n$, whereas Var requires a factor of $n^2$ more sequences to be explored. This is a considerable computational overhead, but, as known from traditional beam search, can be worth the additional compute if downstream performance is absolutely crucial.
> >
> > More broadly, we mostly find VAR to be particularly effective with interleaved reasoning whereas BeamVar prevails in multiple-choice and math questions.
> >
> > **Q: Does stopping on “.” with sketched chain-of-thought prompts restrict the model's reasoning? If no stopping phrase is specified, when does a variable end?**
> > Thank you for this close attention to detail. Indeed, a badly chosen stopping phrase can impact decoding in this way, just like a badly constructed prompt will impact performance. However, upon manual inspection, we don’t find this to be a particular issue in this case. Especially in chain-of-thought prompts, stopping on “.” mostly serves as a proxy for structuring the LLMs response, where LLMs will also recover and continue potentially truncated “thoughts”. Generally, we constructed sketches just like prompts, by developing, testing, and debugging them using real example data (also discussed in 4.3). If no stopping phrase is provided (e.g., for CoT), our decoders continue generation until the model emits the end-of-sequence token on its own.
> >
> > **Q: Why do you shrink VARs $n^2$ generations back down to beam width $n$, before adding the next deterministic chunk rather than extending all $n^2$?**
> > This is an interesting proposal. The reason for our choice is mostly based on cost considerations w.r.t. the OpenAI API. Each additional scored sequence incurs an additional request at the full length of the current sequence with OpenAI. Since `top_n` still maintains the highest scoring continuation, we reduce this cost by focusing on the “most promising” sequences in the current set.  With open/accessible models, however, the proposed alternative would definitely be sensible, considering that scoring a sequence with standard transformers corresponds to a single forward pass only. In our published open source decoder library dclib, this can be easily implemented by moving a single line.
> >
> > We hope to have been able to address all the reviewer’s concerns, are happy to answer any follow-up questions they might have, and are looking forward to their reply.

---

> > > ### Comment · Reviewer_AyRy · 2023-11-23
> > >
> > > Thank you for these detailed responses!
> > >
> > > It seems that the main contribution of the paper is to empirically test whether this kind of guided decoding (variants of which have previously been published in LMQL, Synchromesh, etc. as well as implemented in open-source libraries like Microsoft Guidance and Outlines) help LLMs to reason better. In particular, there appears to be a claim that enforcing relatively simple stylistic constraints on the output ('chains of thought are \n-separated lists beginning with hyphens') improves reasoning.
> > >
> > > But I still do not feel that the paper, or the response, provides satisfying insight into why that is or how generally such a conclusion might hold. As I asked in my review, I am still wondering:
> > >
> > > > Do failures to follow the format cause failures in reasoning?
> > >
> > > For example, if not following a sketch template, how do chain-of-thought-prompted models fail? How often are failures in reasoning correlated with failures to follow the format? Why are these failures of reasoning fixable by forcing them to insert hyphens before each thought? How generally is that true?
> > >
> > > I will not argue against this paper's acceptance if other reviewers want to accept it, but I believe its thesis could be clarified and better defended empirically in future revisions.

---

### Author Response · Authors · 2023-11-21
**Main Response**

We greatly appreciate the Reviewers’ questions, comments, and suggestions and were particularly encouraged to hear that almost all reviewers recognized prompt sketching as a novel (tG6m, pDQp), useful, and intuitive (AyRy, Apoy) approach that identifies new connections between sequence decoding and structured LLM inference (AyRy) and enables novel, practical applications (tG6m).

In response to the reviews, we have extended and improved our manuscript, putting a particular focus on the following two points:
* We want to highlight that our work not only introduces a novel theoretical perspective on template-guided inference and the corresponding decoding procedures but also (for the first time) empirically validates that already existing implementations of simple template-guided (stop-and-go) inference (e.g., Guidance AI [1], LMQL [2]), can indeed improve model reasoning capabilities. Since we believe this was not sufficiently clear, we have improved our description of this contribution in the revised draft.
* We have significantly scaled (up to 10x) and extended many of our experiments to address concerns surrounding our experimental evaluation. These changes include:
   - We have extended our Llama-2 based evaluation to include all decoder/model combinations and observe similar trends as for OpenAI models, but at 10x the scale. In Table 8 (Appendix C.3), we provide confidence intervals for these results.
    - We have scaled our Sudoku and Dungeon Escape experiments by 10x and can confirm our initial findings (sketching significantly improves backtracking and planning capabilities with puzzles and interactive environments).
    - We have added an additional baseline in Appendix D, comparing with self-consistency decoding [4], showing that sketching outperforms self-consistency at a similar compute budget. We note that self-consistency remains orthogonal to sketching, meaning the two approaches can be applied jointly.
    - We have added a new experiment further demonstrating the effectiveness of sketching with interactive LLM-guided exploration of TextWorld [3] environments.

For more details, please see the individual review responses. We are happy to answer any follow-up questions that may arise as part of our responses and look forward to the Reviewers’ response.

**References**
[1] Guidance AI: A guidance language for controlling large language models, S. Lundberg & M. T. C. Ribeiro, https://github.com/guidance-ai/guidance
[2] Prompting is programming: A query language for large language models, L. Beurer-Kellner et al., PLDI’23
[3]  Textworld: A learning environment for text-based games, M.-A. Côté et al., IJCAI’18
[4] Self-consistency improves chain of thought reasoning in language models, X. Wang et al., ICLR’23

---

### Meta-Review · Area_Chair_kkgv · 2023-12-06

**Metareview:**

This paper studies prompting for producing language model outputs that have a desired structure or satisfy constraints. Rather than conditioning on a fixed token sequence and allowing the model to complete it, as with traditional prompting, the authors suggest using sequences broken up into chunks that interleave fixed text and model generated text.  They essentially generalize a technique called stop-and-go inference including allowing it to operate with variations of beam search.

Strengths include some of the experimental results and and the templates that the authors provided, which could prove useful in general. The main weakness here is that the technical contribution is fairly limited, which I explain more below.

I generally felt that the paper is borderline. I ultimately voted for rejecting it for the following reason: most papers on particular prompting techniques tend to be a bit ephemeral since the next iteration of models may well be able to handle what appears to be challenging problems today (i.e., dealing with constraints). Indeed, one of the reviewers makes the same point for the example the authors provided in Figure 1. By itself this is **not** a problem, but it does mean that prompting papers should generally provide something like:
1) very substantial performance improvements versus baselines,
2) a general and new approach that was not there before,
3) an analysis of particular scenarios (domains, distributions, tasks) where particular strategies work and others do not

The current draft didn't quite pass the bar: it doesn't really meet 1), is a generalization of particular techniques so doesn't quite do 2), and only slightly addresses 3). Of course there could be other contributions as well---but as it stands it wasn't quite enough.

With all this said, I'm optimistic for future versions, since the authors did do a bunch of work towards 3) in particular as part of the rebuttal.

**Justification For Why Not Higher Score:**

As described above, the paper's contribution doesn't meet the bar along the axes that prompting papers typically operate on: it doesn't have sufficiently substantial performance improvements, substantial technical innovations, etc.

**Justification For Why Not Lower Score:**

N/A

---

### Decision · Program_Chairs · 2024-01-16

Reject